# Need for cognition moderates the relief of avoiding cognitive effort

**Davide Gheza** [1,2] *, **Wouter Kool**[2], **Gilles Pourtois**[1]

**1** Cognitive and Affective Psychophysiology Laboratory, Department of Experimental Clinical & Health Psychology, Ghent University, Ghent, Belgium, **2** Department of Psychological and Brain Sciences, Washington University in St Louis, St. Louis, MO, United States of America

* gheza@wustl.edu

**Data Availability Statement:** All data and analysis scripts are available from the OSF (osf.io/gwhzf/).

**Funding:** The authors received no specific funding for this work.

## Abstract

When making decisions, humans aim to maximize rewards while minimizing costs. The exertion of mental or physical effort has been proposed to be one those costs, translating into avoidance of behaviors carrying effort demands. This motivational framework also predicts that people should experience positive affect when anticipating demand that is subsequently avoided (i.e., a "relief effect"), but evidence for this prediction is scarce. Here, we follow up on a previous study [1] that provided some initial evidence that people more positively evaluated outcomes if it meant they could avoid performing an additional demanding task. However, the results from this study did not provide conclusive evidence that this effect was driven by effort avoidance. Here, we report two experiments that are able to do this. Participants performed a gambling task, and if they did not receive reward they would have to perform an orthogonal effort task. Prior to the gamble, a cue indicated whether this effort task would be easy or hard. We probed hedonic responses to the reward-related feedback, as well as after the subsequent effort task feedback. Participants reported lower hedonic responses for no-reward outcomes when high vs. low effort was anticipated (and later exerted). They also reported higher hedonic responses for reward outcomes when high vs. low effort was anticipated (and avoided). Importantly, this relief effect was smaller in participants with high need for cognition. These results suggest that avoidance of high effort tasks is rewarding, but that the size off this effect depends on the individual disposition to engage with and expend cognitive effort. They also raise the important question of whether this disposition alters the cost of effort per se, or rather offset this cost during cost-benefit analyses.

## Introduction

Daily life constantly presents us with challenging tasks that we need to perform to achieve our goals. Successful completion of these tasks requires us to invest cognitive effort. Recent theories [2, 3] suggest that such decisions are implemented as simple cost-benefit analyses, where reward is discounted by the anticipated cost of effort. The idea that effort is costly is captured by the 'law of least effort' [4]. This law predicts that, all else being equal, people will tend to avoid effortful actions. There is ample empirical evidence for this principle. For example, Kool

**Competing interests:** The authors have declared that no competing interests exist.

and colleagues [5] demonstrated that human participants prefer choice options with the least cognitive demands (even when time on task and error likelihood were matched). Such effort avoidance has been demonstrated for a wide range of demands, such as response conflict [6, 7], task switching [5, 8], complex task policies [9], and short-term memory load [5]. In fact, a recent study demonstrated that people are willing to endure physical pain to avoid effort [10]. Across these studies, the task demands invoke the need for cognitive control or controlled information processing [11, 12], which are core features in models of mental effort [13]. In short, mental effort carries a cost, and this cost is used to devalue anticipated reward [14–16], resulting in an expected value of control' [2].

If reward is discounted by effort, then it should also be enhanced if effort is anticipated, but eventually avoided. That is, the avoidance of expected effort should be experienced as relief. Interestingly, relief (from anticipated pain) appears to involve the same reward processing systems that underly representations of effort costs [17, 18]. Together with the observation that effort is costly, this predicts that the avoidance of anticipated effort should be experienced as rewarding. To our best knowledge, there is a scarcity of studies that investigate this phenomenon. Here, we report two studies that test this hypothesis.

Earlier work from our group [1] has provided some initial, but ultimately inconclusive, support for this hypothesis. In this experiment, participants performed a well-validated gambling paradigm [19, 20] that was combined with an effort task [21]. On each trial of this task, participants chose between four options that each afforded a chance to win a reward. Critically, on some of the trials, participants were informed that after a loss (a 'no reward' outcome), they would be given the choice to exert effort to repeat the gamble. Therefore, on these 'special' trials, there was a prospect of effort both prior to and during gambling, while for the rest of trials, there was not. Moreover, rewards on special trials were associated with the avoidance of effort (i.e., it rendered the opportunity to spend effort to repeat the gamble moot). Interestingly, the participants reported more positive affect (pleasantness), as well as increased relief for reward delivered on special trials compared to standard trials. In addition, at the electrophysiological level, we found that reward delivered after gambling on special trials elicited a larger reward positivity event-related potential [22] as well as enhanced power in delta [23] and beta-gamma [24–26] frequency bands. Together, these results suggest that relief from effort registers as rewarding [4, 27, 28]. However, an alternative interpretation exists.

To see this, note that people did not just avoid mental effort on rewarded special trials, but they also avoided the extra time spent on deciding whether to repeat the gamble and on the effort task itself. This creates an opportunity cost [29]: special trials with no reward lasted longer, and thus were associated with reduced time to accrue reward. In other words, participants may have experienced more positive affect for rewards on special trials simply because it reduced their time on the experiment. Thus, though these results are suggestive, it remains to be shown whether relief from effort enhances subjective reward.

To test this, we made several key changes to our experimental design [1], so that only demands for mental effort, and not opportunity costs, differed between conditions. In this new task, each trial with no reward was followed by a cognitive effort task [30] performed to recuperate the missed reward. This cognitive effort task required mental arithmetic. At the start of each trial, a cue indicated the amount of demanded effort for the arithmetic task (i.e., easy vs. hard). Comparing self-report ratings between these trials allowed us to isolate the effect of effort avoidance, whereas in our previous study the analogous comparison conflated effort and time. It is worth drawing attention to the fact that each trial with no reward lead to the effort task. Participants did not decide whether or not to perform the task. Therefore, the expectation of effort was high and constant across all trials. These methodological changes allowed us to compare the subjective experience of reward when either an easy or difficult task

was expected, with opportunity cost equal between conditions. In line with our previous study, we assessed the latent construct of the "hedonic response" (i.e., the subjective experience of the gambling trial outcome) using three probes, each addressing a different affective dimension ("pleasure", "frustration", and "relief"). We refer to the interaction of trial outcome with effort anticipation on the hedonic response as a "relief effect"."

The second goal of our study was to assess the source of individual differences in effort-related relief. In addition to objective task demands, valuations of effort are influenced by people's state and trait dispositions. For example, stress increases effort avoidance [31] and depression/anhedonia decreases reward processing [32]. Here, we tested whether relief was predicted by participants' 'need for cognition' (NFC; [33]). The NFC is defined as the tendency to engage in and enjoy effortful cognitive activities (e.g., those requiring thinking and problem solving; see also [34]) and can be conceived as a motivational drive [35]. We hypothesized that an increased NFC would decrease the relief of effort avoidance, and hence behave in an opposite manner compared to stress, depression or negative affect [36]. In other words, the NFC might mitigate the impact of effort (anticipation) on reward processing, with a relatively reduced effort-related relief for participants with a high NFC.

## Methods

### Participants

Twenty-three undergrad students from Ghent University (17 females; median age: 21 years, range: 18–30) participated in Experiment 1. They had normal or corrected-to-normal vision and did not report any history of neurological or psychiatric disorders. Sample size was determined to be at least as large as in our previous experiment [1] where a similar experimental manipulation was used, and where a significant effect of cost anticipation on reward was found (increased pleasantness [p = .008, d = 0.57] and relief [p = .033, d = 0.27]).

Seventy-nine young adults participated in Experiment 2. None of them participated in Experiment 1. To explore individual differences in NFC, the sample size was determined to be as large as available time and resources would allow. Three participants were excluded from further analyses due to low precision in their ratings of the reward-related feedback (see exclusion criteria below). Hence, the final sample consisted of 76 participants (62 females; median age: 21 years, range: 18–46).

These two experiments were part of a more general research project investigating effects of motivation on reward that was approved by the local ethics committee at Ghent University. All participants gave written informed consent prior to the start of the experiment, were debriefed at the end, and received a monetary compensation for their participation. Participants' data was collected between 2018 and 2019 and was anonymized at the time of collection.

### Stimuli and task

For Experiment 1, we adapted a widely used gambling task [1, 37] and combined it with a cognitive effort task [30]. At the start of each trial, participants were informed about the cognitive effort level with a text cue ("easy" vs "hard") located at the center of the screen (1000 ms). Following a fixation dot (1500 ms), four doors appeared on the screen, and participants had to choose one of them by pressing with their left hand the corresponding numeric key (1 to 4) on a keyboard. After another fixation dot (700 ms), this choice was followed by reward-related feedback (1000 ms), indicating either a reward (green "+") of 6 cents, or a no-reward outcome (red "o"). Participants were instructed to guess and select a door containing a reward in order to maximize their payoff. However, unbeknown to them, the outcome was unrelated to the

choice and reward probability was set to exactly 50%. Participants were instructed that in case of a reward, the trial would end and a new trial would follow. Hence, no additional effort would be necessary. However, when there was no reward, a second task would follow, which could be hard or easy (as indicated by the previous cue), so that they could get another chance to win reward. Thus, receiving no reward during the gambling task resulted in the prospect of effort. More specifically, after 1000 ms (fixation), a mental arithmetic task started (i.e., the 'effort task'). This task required participants to complete two calculations (two additions or an addition and a subtraction, all of which with single-digit numbers) (see Fig 1). In the hard condition, every operation required carrying or borrowing. In the easy condition, none of the two operations required carrying or borrowing. This manipulation results in two levels of difficulty, as shown in previous studies [38–40] and confirmed by subjective ratings (see Results below).

The effort task was structured as follows: a pound symbol indicated the start (400ms); digits and arithmetic signs were then presented serially, each lasting 500ms and interleaved with blank screens (200ms in Experiment 1, 100ms in Experiment 2); finally, two possible solutions were presented simultaneously, and the participants had to choose the correct one by pressing the corresponding key with the right hand (i.e. numeric keypad; "1" for the leftmost or "2" for the rightmost solution). They were instructed to select the correct answer as quickly as possible, with a time limit of 4000ms. After this choice, a blank slide was presented (1000 ms), followed by a new feedback screen related to their performance (1000 ms): a reward outcome (green "+") indicated a correct response and a win of 6 cents, and a no-reward outcome for incorrect responses (red "o") was presented. If participants missed their response, or if they responded too late, a screen indicated that there was "no response detected". Serial

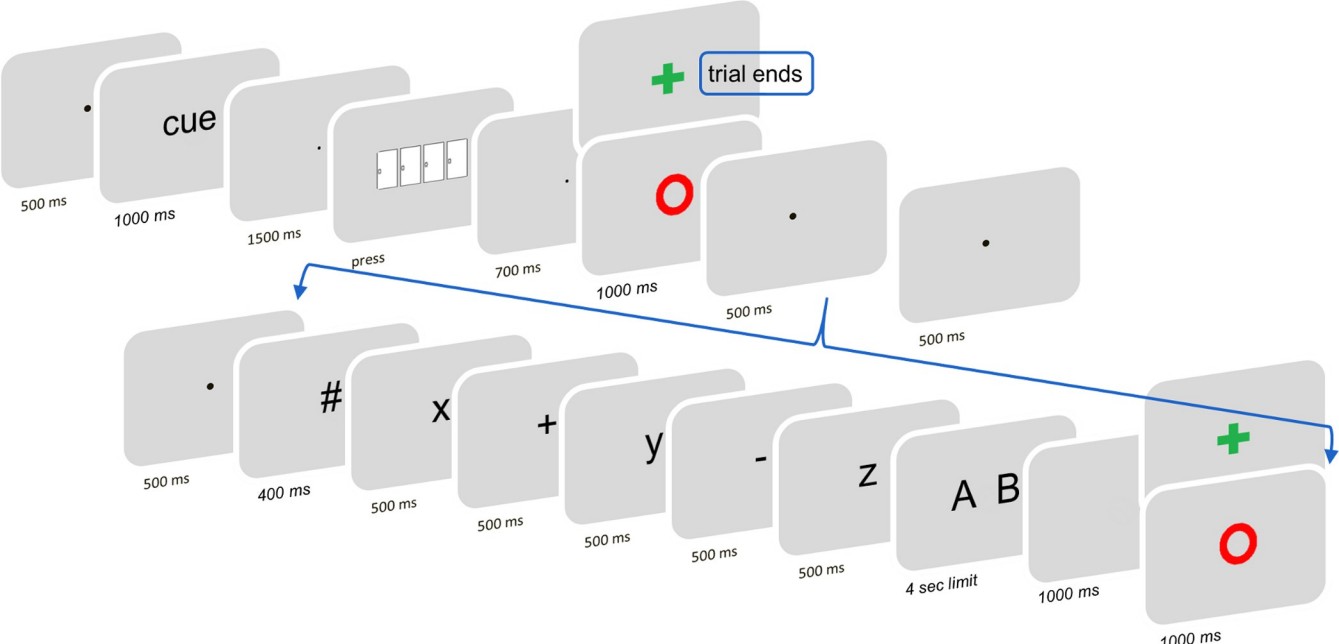

**Fig 1. Overview of the task and trial structure.** Participants were first informed about the cognitive effort level with a text cue (word "easy" or "hard"). After they picked one door, they received reward or no-reward feedback (50% reward probability). Only in case of no-reward feedback, the effort task ensued. For both conditions, the initiation of the effort task was probed by an uninformative pound symbol (#), followed by a serially-presented list of digits and operations. Participants were instructed to integrate them, and based on this arithmetic select the correct one out of two possible answers. Based on their answer, "performance" feedback was given, providing reward in case of a correct answer. On a small proportion of trials, the subjective hedonic value of the reward-related feedback was assessed with probes presented 1000 ms after its offset.

presentation of digits and operations was chosen in order to equate the time spent observing stimuli between conditions, as well as to avoid calculation strategies [30]. Across trials, different combinations of digits and arithmetic signs were used to avoid learning or habituation. After the effort task, a new trial of the gambling task followed. The intertrial interval was fixed and set to 1000 ms.

The subjective value of the first feedback screen (after gambling) was assessed by specific probes. In a few trials (n = 48), 1000 ms after the offset of the gambling outcome, three questions were presented, probing the perceived pleasantness, frustration, and relief of the outcome. Participants answered them using visual analog scales (VAS). These three ratings were submitted 12 times for each effort level (low vs. high) and outcome (reward vs. no reward) condition. By using three questions, which assessed a range of different affective dimensions, we aimed at measuring the latent construct of a 'hedonic response' along a negative-positive valence axis [41]. Multiple self-report items allowed us to validate the consistency of participants' responses, and to increase statistical power by controlling for nuisance variability between affective dimensions.

Experiment 1 consisted of 208 trials, including an equal amount of easy and hard trials. The gambling task had a pre-set reward probability of 50%, and the effort task had to be completed only in case the choice did not deliver reward. Therefore, the effort task was administered 52 times for each difficulty level.

Seven self-paced breaks were distributed equally throughout the experiment. At the end of each break, participants were asked to rate the difficulty of the effort task, its pleasantness, their motivation to complete it, and their satisfaction with correct performance on the task. Each question was submitted twice, for each of the two difficulty levels, using a VAS. Participants received a fixed €8 compensation for their participation. Depending on their accuracy with the effort task, a maximum payoff of €12.48 could be earned (mean = €11.91).

For Experiment 2, where the NFC questionnaire was also administered [42], the same procedure was used but a few changes were made. The reward feedback with the gambling task indicated that 5 cents were won (instead of 6). The three ratings about the gambling outcome were submitted each 40 times in total (instead of 48), 1000ms after its offset (10 times for each combination of difficulty level and outcome). In Experiment 2 we also probed the subjective value of the performance feedback (after the effort task). Analogous to the reward-related feedback, three ratings probing the perceived pleasantness, frustration, and relief of this feedback. These were presented directly after the performance feedback. Performance feedback ratings were submitted up to 26 times, equally split between high and low effort, but were omitted in case of an incorrect response. At this stage we only probed affective ratings for correct feedback as the effort task was tuned to induce only very few errors. Reward-related feedback (gambling task) and performance feedback (effort task) were never probed in the same trial. Experiment 2 consisted of 200 trials, with an equal amount of easy and hard trials (100 each). For each of these, there were 50 trials where the effort task had to be performed (no reward) and 50 trials where it was avoided (reward). Participants were compensated with €8 for their participation. Depending on their accuracy with the effort task, they could earn up to €10 (mean = €9.63). Last, in Experiment 2 we added 16 "catch trials" to promote and assess participant's attention to the effort cues presented at the beginning of the trial. More precisely, after door selection participants were asked to report how hard a following effort task would be. After their response, participants received catch-related feedback (correct, incorrect, or too late response) and the catch trial terminated. Eight catch trials, for each of the two effort levels, were randomly interspersed among the 200 task trials.

The experiments' duration was approximately 60 minutes, including instructions and a short practice. In Experiment 2, the NFC questionnaire was administered after the task. All

stimuli were shown against a grey homogenous background on a 21 inch CRT screen and controlled using E-Prime [43].

## Data analysis

We evaluated the effectiveness of the cognitive effort manipulation by comparing performance on the effort task (i.e., accuracy and speed) between the easy and hard conditions. Moreover, we also compared their subjective value by analyzing the ratings of difficulty, pleasantness, motivation to perform well, and pleasure in performing well. These ratings were first transformed to percentages, setting anchors to the boundaries of the scales, and were averaged across the seven repetitions.

The subjective ratings of the reward-related feedback obtained for each difficulty level (easy vs. hard), outcome (reward vs. no reward), and affective dimension (pleasantness, frustration, and relief) were also first transformed to percentages, setting anchors to the boundaries of these scales. VAS scores were calculated by first identifying the relative x-axis position of the mouse with respect to the leftmost position of the scale at the time of participant's click. We then converted this measure from pixels distance to percentage over the scale range (396 pixels). For the "frustration" scale, we reverse-scored the percentages in order to provide comparable ratings for the three affective dimensions. Subjective ratings for the three affective dimensions were used to assess the hedonic response as a whole, along a negative-positive valence axis. In all analyses, we predicted single trial subjective ratings with a mixed modelling approach, in which participant and affective dimension were treated as random effects (i.e., considered nuisance factors).

For Experiment 2, subjective ratings of the performance feedback were analyzed in the same way. Additionally, for statistical analyses the VAS scores and NFC data were centered by subtracting the group means from the individual scores.

Participant's data were excluded from further analysis if the average mouse-click x-coordinate of the reward-related feedback rating was above 105% or below -5% of the rating range, in any of the 3 affective dimensions, and any of the 4 levels of outcome by difficulty.

## Statistical analyses

Subjective ratings data were analyzed using Bayesian model comparison. Inference about their generative processes was based upon Bayes Factors (BFs), computed for alternative explanatory models in ANOVA designs ([44], see also [45]). The analyses' pipeline was implemented in R v4.0.5 [46] with the package BayesFactor v0.9.12–4.2 [47], and involved: I) defining theoretically sound probability models; II) computing BFs, i.e. the ratio between the likelihood of each model of interest (the probability of the observed data, given the model/hypothesis) and the likelihood of the null model; III) model selection based on the highest BF; IV) characterizing the direction of follow-up contrasts by means of Bayesian t-test between conditions of interest. The models' likelihood was estimated using Markov-Chain Monte Carlo simulations with 10,000 iterations, and BFs were computed assuming a wide Cauchy prior centered on zero: $d \sim Cauchy (0, 0.707)$.

For the gambling task feedback ratings, used to estimate the underlying hedonic response, the models of interest included the effects of 1) *outcome*, 2) *difficulty level*, 3) *outcome + difficulty level*, and 4) *outcome x difficulty level*. Additionally, for Experiment 2 we tested a model including 5) the three-way interaction of outcome x difficulty level x NFC score. Moreover, the factors Subject, Affective dimension, and their interaction were treated as nuisance variables (i.e., varying effects to control for). To do this, they were entered in all models, including the null model. Additionally, the null model included a single (i.e., constant) intercept. This

procedure allowed us to control for the nuisance effects of 1) individual differences in VAS biases (e.g., preference for a given section of the scale), 2) overall differences in rating across affective dimensions (e.g., overall higher responses for a specific scale), and 3) individual variations in the latter differences.

In Experiment 2, to further explore the effect of NFC on reward processing, we also ran a full Bayesian estimation of parameter values for model 5, using the "brms" R package [48]. To analyze performance feedback ratings (Experiment 2), we used a one-tailed Bayesian t-test to estimate the evidence in favor of increased positive evaluation of a reward outcome after high vs. low effort, as compared to a null model.

## Results

### Experiment 1

Accuracy on the effort task was higher for the easy (M = 98%, SD = 14) compared to the hard condition (M = 87%, SD = 34; BF+0 = 2.20 x $10^3$). Mean reaction time was larger for the hard (M = 1091 ms, SD = 476) compared to the easy condition (M = 621 ms, SD = 144; BF+0 = 4.11 x $10^3$). These results indicated that the difficulty manipulation was successful.

The subjective ratings of the effort task indicated these two difficulty levels were experienced as clearly different. The hard compared to easy condition was perceived as more difficult (M easy = 6.1, SD = 7.3; M hard = 31.2, SD = 18.7; BF-0 = 1.09 x $10^5$) and less pleasant (M easy = 83.4, SD = 12.5; M hard = 62.7, SD = 21.0; BF+0 = 1.92 x $10^3$), while participants reported similar levels of motivation to perform them correctly (M easy = 85.1, SD = 14.8; M hard = 86.6, SD = 13.6; BF01 = 3.58), as well as pleasure in performing them correctly (M easy = 81.8, SD = 17.1; M hard = 76.4, SD = 18.5; BF01 = 2.00).

The hedonic responses to the gambling outcome (i.e., reward-related feedback ratings; Fig 2) were best explained by a model that contained the *outcome* x *difficulty level* interaction. Under this model the observed data were BF10 = 5.28 x $10^{648}$ times more likely to be produced than under the null model. Moreover, the *outcome* x *difficulty* model explained the observed data 284 times better than the second-best model, which only included the main effect of *outcome* (Table 1). Follow-up Bayesian one-tailed t-tests showed strong evidence for the hypothesis that no-reward outcomes were evaluated as more positive in low-effort compared to high-effort trials (BF+0 = 6.00 x $10^5$). Conversely, for reward outcomes, there was only weak support (BF+0 = 2.22) for more positive evaluations in the high- compared to low-effort trials. In other words, participants rated a no reward outcome as more positive when they anticipated low effort.

As can be seen from the S1-S6 Figs in S1 File, participants showed a robust internal consistency in their rating of the reward-related feedback across outcome levels: Their rating was consistently lower for no-reward compared to reward outcome. Moreover, we observed very robust correlations between pairs of affective dimensions (group-level statistics of these correlations are reported in the S1 File).

### Experiment 2

Participants paid attention to the effort cues presented at the beginning of each trial, as suggested by high accuracy in reporting them during catch trials (M easy = 96.2, SD = 8.9; M hard = 91.8, SD = 13.8).

Similarly to Experiment 1, the accuracy for the effort task was higher for the easy (M = 99%, SD = 12) compared to the hard condition (M = 86%, SD = 34; BF+0 = 3.23 x $10^{16}$). Mean reaction time was larger for the hard (M = 1267 ms, SD = 467) compared to the easy condition

## Reward feedback

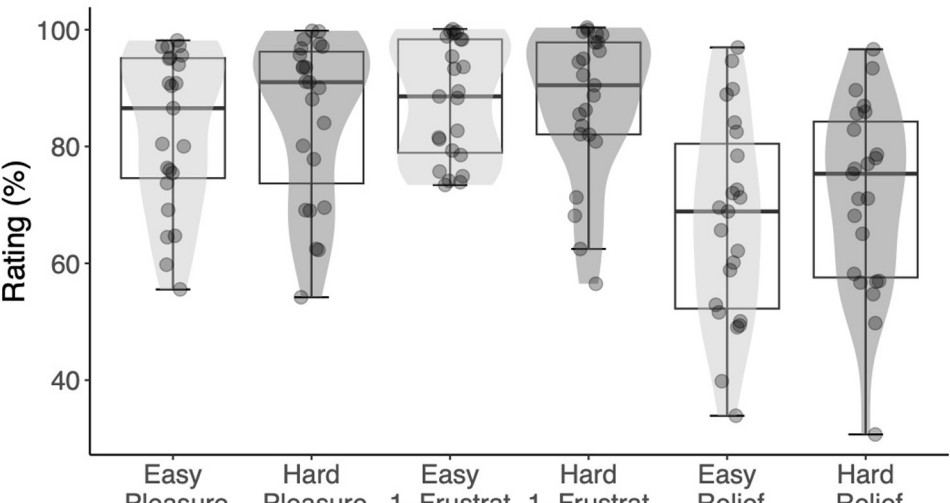

## No-reward feedback

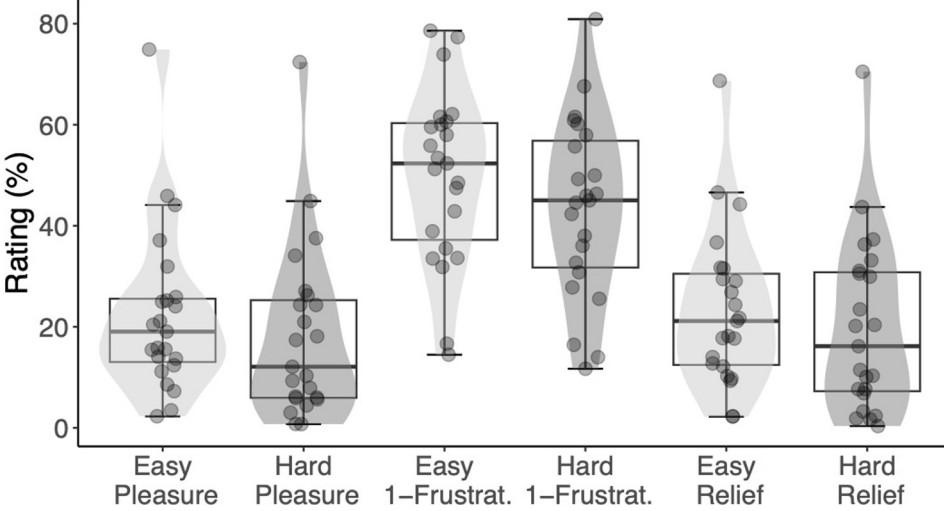

**Fig 2. Experiment 1.** Ratings of reward-related feedback that followed the gambling task, by difficulty level (easy or hard), and affective dimension (pleasure, reverse-scored frustration, and relief). Each dot represents the subject-level average across 12 repetitions of the rating. Reponses to the three affective dimensions were used to estimate the underlying hedonic response along a single valence axis. For either a reward (top) or no-reward (bottom) outcome, light grey density shades correspond to low effort anticipation and dark grey density shades correspond to high effort anticipation. The horizontal line represents the median, the box represents the interquartile range, and the whiskers extend to the last data point within 1.5 times the interquartile range.

(M = 694 ms, SD = 198; BF+0 = 7.21 x $10^{19}$). This demonstrated that the difficulty manipulation was successful.

The subjective ratings of the effort task again indicated these two difficulty levels were experienced as clearly different. The hard compared to easy condition was perceived as more difficult (M easy = 14.0, SD = 15.7; M hard = 40.2, SD = 23.1; BF-0 = 4.39 x $10^{14}$) and less pleasant

**Table 1. Bayesian model comparison (Experiment 1).**

| Model | BF10 | % pe |
|---|---:|---:|
| [1] Subject * AffDim + efflev | 0.05 | ±3.39% |
| [2] Subject * AffDim + outcome + difflev | 2.33 x 10^645 | ±2.58% |
| [3] Subject * AffDim + outcome | 1.85 x 10^646 | ±2.22% |
| [4] Subject * AffDim + outcome * difflev | 5.28 x 10^648 | ±4.32% |
| Against denominator: | | |
| percent ~ 1 + Subject * AffDim | | |

*Note*. Bayes factors ($BF_{10}$) and percentage of proportional errors (% pe) for each model relative to the null. The models are sorted from top to bottom with ascending Bayes Factor.

(M easy = 76.7, SD = 19.9; M hard = 57.0, SD = 21.7; BF+0 = 6.46 x $10^7$). In line with Experiment 1, participants reported similar levels of motivation to perform the easy and hard tasks correctly (M easy = 82.0, SD = 16.1; M hard = 84.5, SD = 15.7; BF01 = 2.01), and also similar pleasure in performing them correctly (M easy = 76.3, SD = 19.3; M hard = 79.1, SD = 17.8; BF01 = 3.39).

Next, we turned our attention to the analyses of hedonic responses as a function of both gambling outcome and difficulty (Fig 3). Those ratings were again best explained by a model that included the *outcome* x *difficulty level* interaction. Under this model, the observed data were BF10 = 7.52 x $10^{1209}$ times more likely to be produced than under the *null* model. This model also explained the observed data 2.11 x $10^{20}$ times better than the second-best model, which only included the main effects of *outcome* and *difficulty level*. The follow-up Bayesian one-tailed t-tests showed strong evidence for the hypothesis that no-reward outcomes were evaluated as more positive on low-effort compared to high-effort trials (BF+0 = 1.55 x $10^{14}$). Conversely, for reward outcomes, there was strong evidence that evaluations were more positive in the high compared to low effort trials (BF+0 = 2.08 x $10^5$). In sum, we replicated Experiment 1: participants rated the no reward outcomes as more positive when they anticipated low effort. In extension of Experiment 1 (Fig 2), participants also rated the reward outcomes as more positive when they anticipated high effort (Fig 3). This result suggests that participants experienced relief when high effort was anticipated but avoided.

To test the hypothesis that a predisposition towards cognitive effort (as measured with the NFC questionnaire) would reduce this relief effect, we included participants' NFC score as a continuous predictor in the best-fitting model described above (the two-way model including the interaction between outcome and difficulty level). Under this new three-way interaction model (Table 2), observed data were more likely compared to the former best model (BF = 2.05 x $10^{12}$) and the null model (BF = 1.54 x $10^{1222}$). In other words, there was strong evidence that the disposition towards cognitive effort moderated the interaction effect between outcome and difficulty level.

Furthermore, when selectively predicting the feedback ratings for the 'relief' affective dimension, data were still best explained by a model that included the three-way interaction of *outcome* x *difficulty level* x *NFC*, followed by a model including the two-way *outcome* x *difficulty level* interaction (see S1 File).

To further explore the effect of NFC on reward processing, we conducted a full Bayesian estimation of parameter values. We fit model 5) with the "brms" R package [48], and we inspected the posterior probability distributions for every interaction parameter that included NFC. In Table 3, we report all the estimated model parameters. Confirming the previous model comparison, we observed a negative three-way interaction *outcome* x *difficulty level* x

## Reward feedback

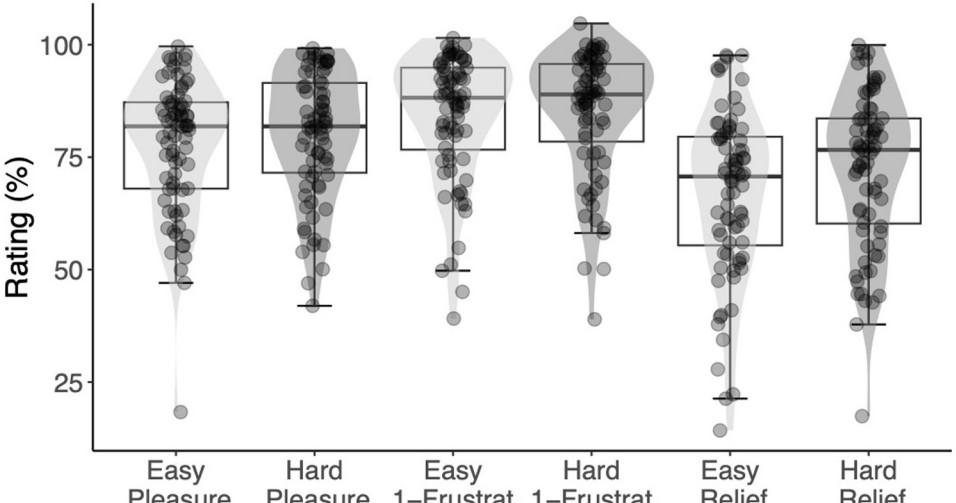

## No–reward feedback

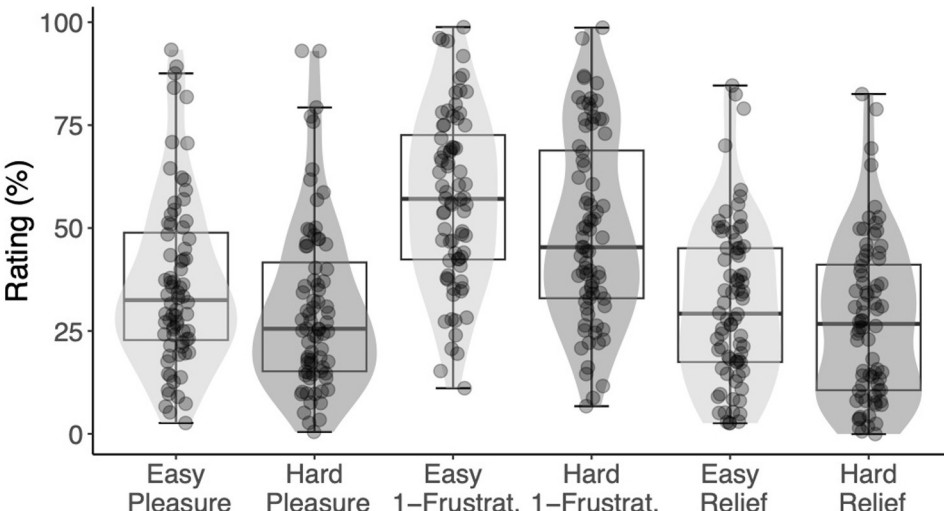

**Fig 3. Experiment 2.** Ratings of reward-related feedback that followed the gambling task, by difficulty level (easy or hard), and affective dimension (pleasure, reverse-scored frustration, and relief). Each dot represents the subject-level average across 10 repetitions of the rating. Reponses to the three affective dimensions were used to estimate the underlying hedonic response along a single valence axis. For either a reward (top) or no-reward (bottom) outcome, light grey density shades correspond to low effort anticipation and dark grey density shades correspond to high effort anticipation. The horizontal line represents the median, the box represents the interquartile range, and the whiskers extend to the last data point within 1.5 times the interquartile range.

*NFC* (estimate = -0.15; 95% credible interval = [-0.27 0.04]; posterior probability = 0.98), indicating that the *outcome* x *difficulty level* interaction was moderated by *NFC*. As can be seen in Fig 4, the differences in subjective ratings between difficulty conditions were attenuated, in both reward conditions, for participants with higher NFC scores. This effect can also be observed in the raw data in Fig 5, where the *outcome* x *difficulty level* crossover interaction is

**Table 2. Bayesian model comparison (Experiment 2).**

| Model | BF10 | % pe |
|---|---|---|
| [1] Subject * AffDim + efflev | 0.20 | ±3.92% |
| [2] Subject * AffDim + outcome | 2.22 x 10^1189 | ±3.89% |
| [3] Subject * AffDim + outcome + difflev | 3.55 x 10^1189 | ±4.37% |
| [4] Subject * AffDim + outcome * difflev + NFC | 1.07 x 10^1209 | ±4.36% |
| [5] Subject * AffDim + outcome * difflev | 7.52 x 10^1209 | ±4.53% |
| [6] Subject * AffDim + outcome * difflev * NFC | 1.54 x 10^1222 | ±9.71% |
| Against denominator: | | |
| percent ~ 1 + Subject * AffDim | | |

*Note.* Bayes factors ($BF_{10}$) and percentage of proportional errors (% pe) for each model relative to the null. The models are sorted from top to bottom with ascending Bayes Factor.

shown for two subsamples of participants scoring at the extremes of the NFC scale (participants within the 0–30 percentiles were included in the Low NFC group; participants within the 70–100 percentiles were included in the High NFC group). Both the more positive ratings of reward feedback in the difficult condition, as well as the more positive ratings of no reward feedback in the easy condition were attenuated for people in high in NFC.

Additionally, we observed a negative two-way interaction *outcome* x *NFC* (estimate = -0.24; 95% credible interval = [-0.32–0.16]; posterior probability = 1), suggesting that the main effect of outcome was also moderated by *NFC*. As can be seen in Fig 4, participants low on NFC reported more extreme ratings for both reward and no-reward feedback.

To investigate this pattern of results further, we also verified whether NFC predicted cognitive performance in the arithmetic task. We analyzed reaction times and accuracy of the arithmetic task's responses and compared models including the effects of 1) *difficulty level*, 2) *NFC*, 3) *difficulty level + NFC*, 4) *difficulty level x NFC*. The null model was a simple intercept model, and all models included the random factors *participants* as nuisance. Reaction times (for correct responses) were best explained by the *difficulty level X NFC* interaction model, under which the observed data were BF10 = $1.62 \times 10^{336}$ times more likely to be produced than under the *null* model. This model also explained the observed data BF10 = 8.22 times better than the second-best model, which only included the main effect of *difficulty level*. Similarly, accuracy was best explained by the *difficulty level X NFC* interaction model, under which the observed data were

**Table 3. Posterior parameter estimation summary (Experiment 2).**

| Parameter | Median | 95% CI | pd | Rhat | ESS |
|---|---|---|---|---|---|
| (Intercept) | -15.56 | [-18.78–12.33] | 100% | 1.002 | 1137 |
| outcomereward | 34.36 | [33.11 35.60] | 100% | 1 | 11017 |
| difflevhigh | -5.56 | [-6.80–4.31] | 100% | 1 | 11471 |
| NFC | 0.07 | [-0.11 0.27] | 77.24% | 1 | 2447 |
| outcomereward : difflevhigh | 8.57 | [6.80 10.34] | 100% | 1 | 10051 |
| outcomereward : NFC | -0.24 | [-0.34–0.14] | 100% | 1 | 10621 |
| difflevhigh : NFC | 0.09 | [-0.01 0.19] | 95.76% | 1 | 10693 |
| outcomereward : difflevhigh : NFC | -0.15 | [-0.29–0.01] | 98.47% | 1 | 9637 |

*Note.* Model 5 parameter estimation using the R package "brms". Summarized posterior distributions for constant effects, presenting median values, 95% Credible Interval, probability of direction (pd; proportion of posterior samples with the same sign as the median). Rhat and Effective sample size (ESS) are diagnostic metrics (at convergence, Rhat = 1). Factors are dummy coded.

## outcome x difficulty level x NFC

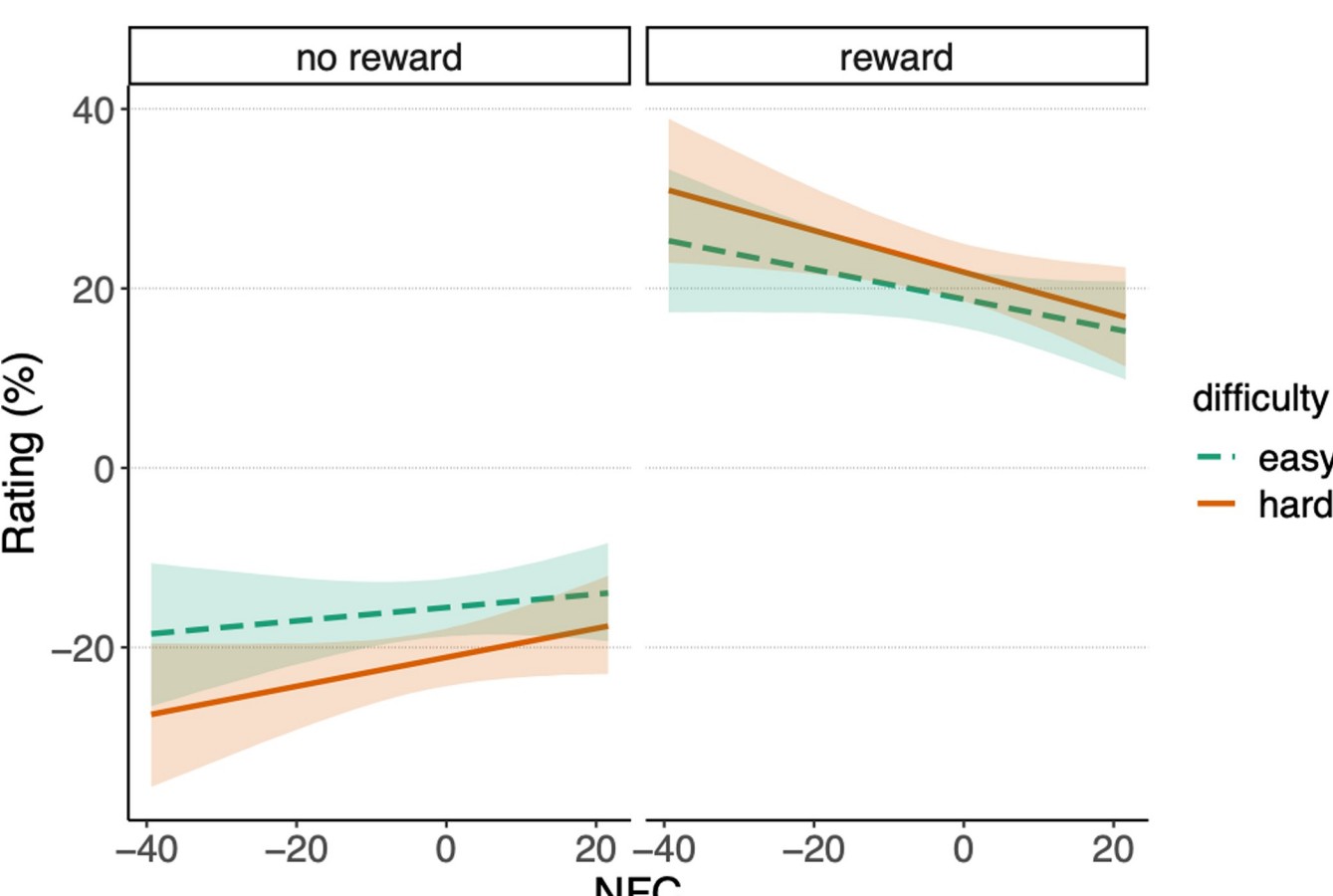

**Fig 4. Results of the Bayesian parameter estimation for model 5.** Interaction effects of predictors outcome, difficulty level, and NFC on the estimated feedback rating. The regression lines represent the mean of posterior probability samples, for each condition and NFC score. The shading represents the 95% credible interval around them.

BF10 = 3.47 x $10^{74}$ times more likely to be produced than under the *null* model. This model explained the observed data BF10 = 8.59 times better than the second-best model, which only included the main effect of *difficulty level*. In other words, for both reaction times and accuracy, there was moderate evidence in support of the hypothesis that NFC interacted with the difficulty level in predicting performance. As can be seen in Fig 6, while the two subsamples of participants scoring at the extremes of the NFC scale showed similar performance for the easy condition, participants scoring high on NFC were more accurate and faster in the hard condition.

Finally, we analyzed the subjective ratings of the performance feedback (Fig 7). Participants rated positive feedback following the high effort task as more positive compared to following the low effort task. The Bayesian one-tailed t-tests showed strong evidence for this hypothesis (M hard = 74.8, SD = 23.2; M easy = 71.4, SD = 24.3; BF+0 = 6.99 x $10^4$).

## Discussion

Exerting cognitive effort carries a cost [50]. Over the last decade there has been an abundance of empirical evidence that supports this notion, showing that effort exertion is avoided [5, 7, 8] or traded for reward. Here, by extension, we hypothesized that the avoidance of anticipated

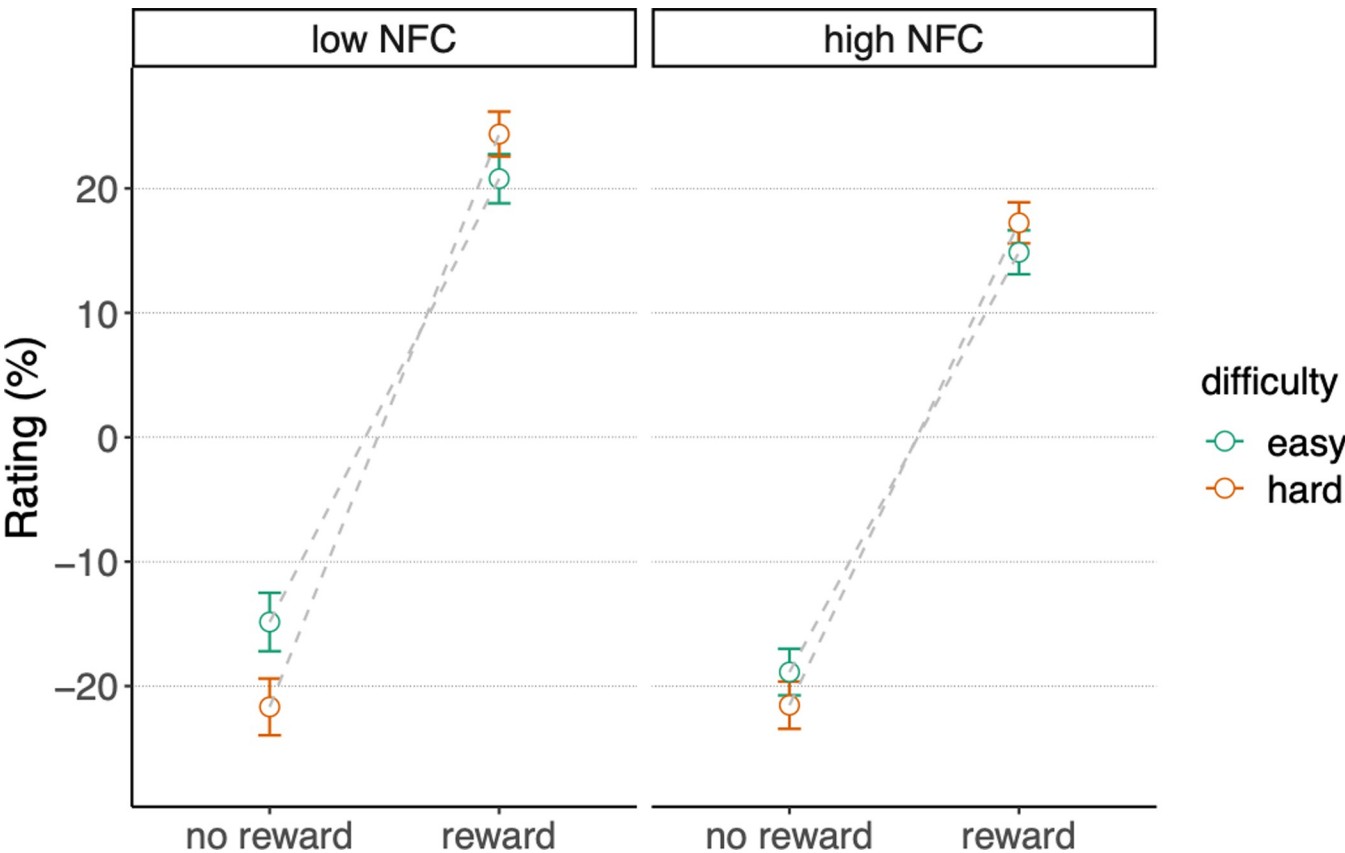

**Fig 5. Ratings of reward-related feedback that followed the gambling task.** Crossover interaction between outcome and difficulty level for two subsamples of participants with extreme NFC scores. Low NFC = participants within 0–30 percentiles. High NFC = participants within 70–100 percentiles. The three affective dimensions (pleasure, reverse-scored frustration, relief) and the ten repetitions were aggregated. Error bars represent the within-subject confidence intervals. The inter-subject variance has been removed by subtracting the subject mean from each rating, and adding the group mean [49].

cognitive effort carries positive value, because it provides relief from an anticipated cost, which in turn bolsters reward [1, 51–57]. We tested this hypothesis by adapting a paradigm that we previously used to show that rewards were perceived as more positive when they signaled avoidance of a demanding task [1]. However, in this previous experiment, avoidance of the demanding task meant that no task would be performed at all. Therefore, this result could also be driven by opportunity costs [29] rather than effort per se. Here, we adapted this experimental paradigm to overcome this limitation. In our new task, we measured subjective ratings to estimate hedonic responses to outcomes that signaled whether an upcoming arithmetic task [30] could be avoided. On each trial, this arithmetic task carried either low or high effort demands, which allowed us to specifically measure the effect of effort context on ratings. In addition, we ran a replication of this study which also investigated whether NFC [33] would act as a moderator on any relief effects. A number of important new results emerged from this study.

First, in both Experiments 1 and 2, the performance measures and subjective ratings suggested that people found the effortful version of the arithmetic task to be more demanding (replicating Vassena et al. [30]). Participants were slower and made more errors on the hard task, and rated it as more difficult and less pleasant than the easy one. However, at the same time, the motivation to perform both task's versions was high and similar, and participants also reported the same amount of pleasure when successfully completing it. These results

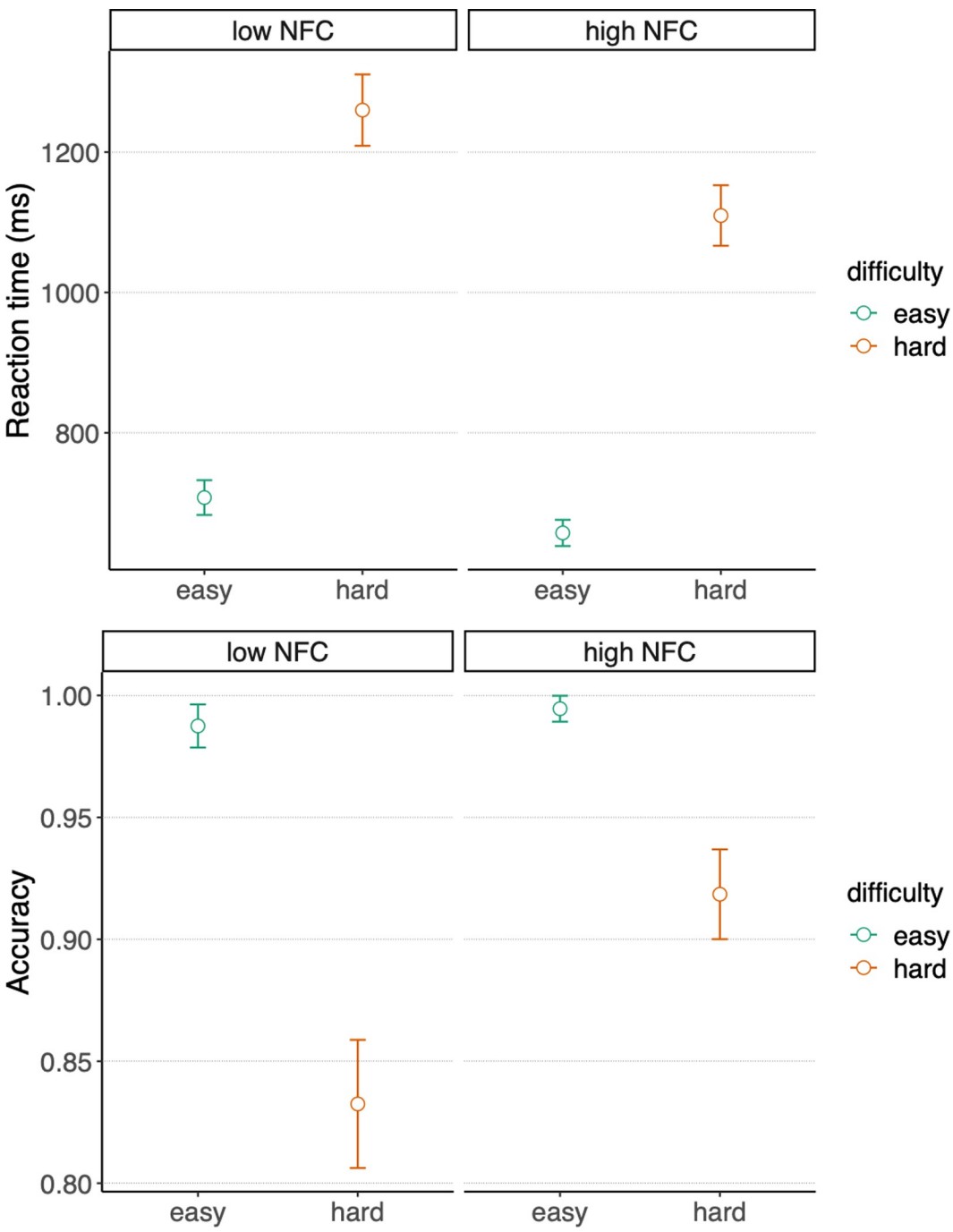

**Fig 6. Effect of NFC on arithmetic performance.** Reaction times (top) and accuracy (bottom) are reported for two subgroups of participants scoring below the 30th percentile (low NFC, N = 20, NFC < 33) or above the 70th percentile (high NFC, N = 23, NFC > 46.5) of the group distribution of NFC scores, separately for the easy and the hard arithmetic conditions. Error bars represent the within-subject confidence intervals. The inter-subject variance has been removed by subtracting the subject mean from each rating, and adding the group mean [49].

suggest that the hard task yielded more negative ratings because of the costs associated with increased effort exertion [36], yet this negative evaluation did not impinge on the participants' motivation. We believe that our data shows that participants exerted more cognitive effort in

## Correct performance feedback

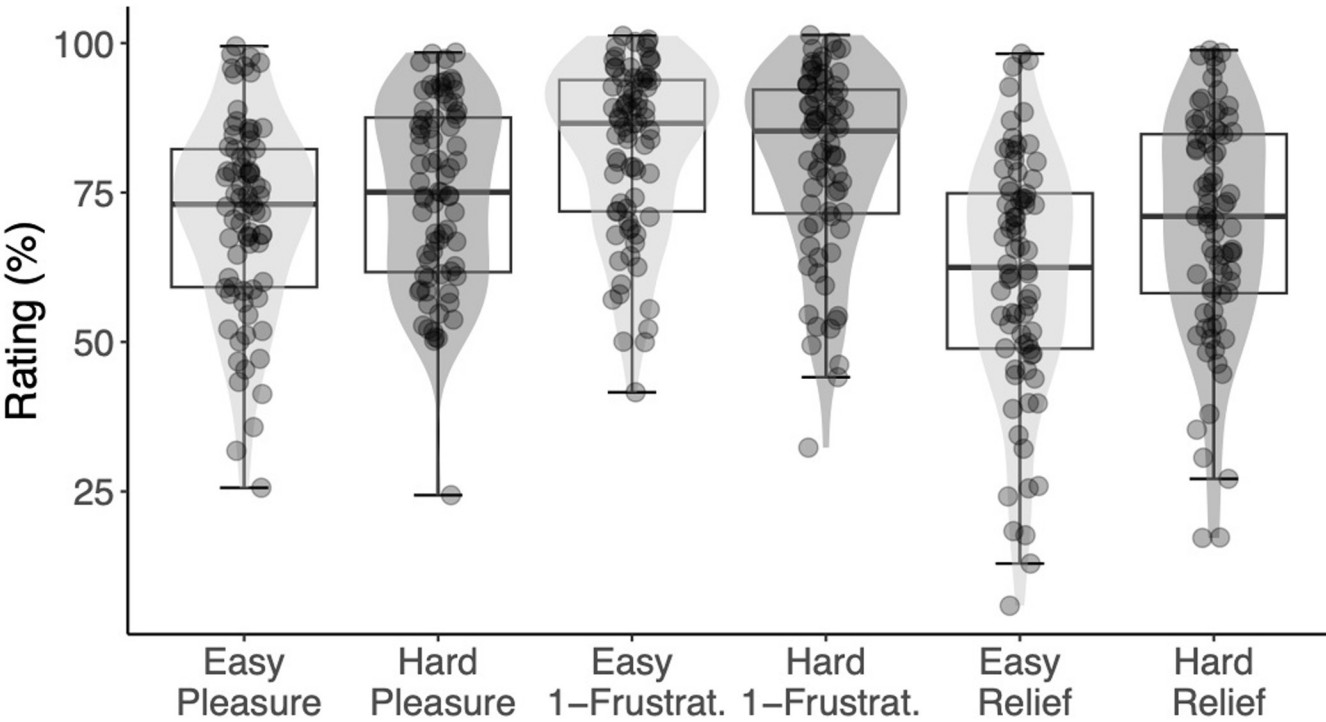

**Fig 7. Ratings of performance feedback that followed the effort task, by difficulty level (easy or hard), and affective dimension (pleasure, reverse-scored frustration, and relief).** Each dot represents the subject-level average across ~13 repetitions of the rating. Reponses to the three affective dimensions were used to estimate the underlying hedonic response along a single valence axis. Light grey density shades correspond to correct feedback following low effort and dark grey density shades corresponds to correct feedback following high effort. The horizontal line represents the median, the box represents the interquartile range, and the whiskers extend to the last data point within 1.5 times the interquartile range.

the hard condition. In response to a substantial difference in difficulty, they only showed a relatively minor drop in accuracy. Most importantly, we saw no clear shift in reported motivation. Nevertheless, it is important to note that our findings provide only preliminary support for this conclusion, as both task demands and the precision of the information-processing operations (i.e., performance) differed, to some extent, between difficulty conditions [13].

Second and central to our main hypothesis, we found that rewards were perceived as more positive when they meant avoidance of the harder, more effort-demanding, task. This results extends our prior findings [1], but now more clearly indicating that the avoidance of cognitive effort per se is perceived as rewarding, translating to a form of relief. Indeed, previous neuroscientific research suggests that the relief of pain involves reward processing pathways [17, 18]. Our finding also suggests that reward is evaluated in relation to prospective effort. Importantly, because we ensured that opportunity costs [29] were equal between effort conditions, we can be sure that this relief effect is truly driven by effort anticipation.

In both experiments, we also found that effort anticipation influenced ratings of unrewarded outcomes, which signaled that effort could not be avoided. Specifically, people viewed no reward outcomes as more positive if they led to a low- compared to a high-effort arithmetic trial. This result suggests that the prospect of exerting low amounts of effort is perceived as more positive compared to exerting high amounts of effort. This finding is compatible with

earlier empirical studies and theoretical models that have associated effort with avoidance and/ or reward devaluation/discount (e.g., [2, 5, 58, 59]).

Interestingly, in Experiment 2 we found that this relief effect was modulated by participants' NFC [33]. As expected, people with high NFC showed a smaller relief effect. This finding is important because it shows that the cost-benefit analyses carried out during effort allocation do not just consider structural factors (such as reward probability or objective task difficulty), but also specific dispositions or motivational states [58]. Therefore, our results inform theoretical models of metacontrol, and of the role of motivation in decision making [60]. They clearly signal the need for such models to consider intrinsic disposition towards effortful tasks, and show how they may influence the processing of reward (see [61]).

It is tempting to think that the attenuated relief effect for individuals high in NFC suggests that such individuals are not averse to cognitive effort, or perhaps even value its exertion [58]. This would pose a challenge to the, presumably universal, law of least effort [5]. In fact, recent research suggests that explicitly rewarding the selection of high-effort actions biases people to choose more effortful lines of action [62, 63]. However, an increased NFC may simply reflect a heightened subjective value of succeeding at demanding tasks, or of the rewards obtained after demanding tasks. Alternatively, people with high NFC may believe that completing hard tasks demonstrates self-efficacy or competence. Under this view, these factors would offset the intrinsic cost of effort during decision making [64]. In other words, a NFC may reflect heightened stakes for cognitive success, rather than lower cognitive costs. Future research may specifically address these open questions by probing in a more granular fashion which of these factors affect effort-based decision making. Finally, we hasten to mention that people high on NFC showed an attenuated relief effect, but not a reversed one. This finding is not inconsistent with the idea that these people simply carry a smaller, but still positive, cost of mental effort.

Interestingly, the results from Experiment 2 also revealed that participants high on NFC solved the hard arithmetic task faster and better than those low on this dimension. This result appears to be at odds with previous studies available in the literature. A range of studies have not found any systematic relationship between the NFC and behavioral performance (using a variety of tasks that demand cognitive control functions such as conflict processing or response inhibition; [65]). Accordingly, some caution is needed when interpreting the modulatory influence of the NFC on the relief effect. While it is possible that this effect is driven by an increased intrinsic motivation to engage in effortful tasks (as explained above), it is also possible that it was caused by the level of required effort being "objectively" lower for participants high on NFC. In other words, the increased task performance on the hard arithmetic task by this group of participants may reflect that it was not difficult enough for them, leading to an attenuated relief effect. Another possibility is that participants scoring low in NFC, and performing sub-par in the arithmetic task, may have developed lower expected values for high difficulty trials, corresponding to lower estimates of reward probability. Therefore, the relief for avoiding high effort may be compounded by the relief for avoiding risky arithmetic trials. To tentatively rule out these hypotheses, in the Supporting Information we report two control analyses which confirm the relief effect, and the role of NFC in moderating it, even when controlling for individual differences in arithmetic performance.

However, because individual differences in performance may index relative differences in expected effort between conditions, our experimental design is not suited to ultimately adjudicate between these competing, but not mutually exclusive, accounts. Future research may address this issue with different experimental procedures, for example by omitting performance feedback. That being said, we would like to note that the questionnaire was administered after task completion, so it is not possible that it induced biases or expectations about effort exertion that influenced behavior on the task. Rather, it is possible that experiencing the

task may have biased participants' responses in the NFC questionnaire. Perhaps, observing their performance on the task influenced the self-evaluations prompted by the NFC question-naire, leading their scores to reflect a combination of stable trait features with transient affec-tive reactions. More generally, the correlational nature of this finding prevents us from drawing any causal inferences between the relief effect and NFC. Hence, additional studies are needed to determine the mechanistic nature of the modulation of the relief effect by NFC. For example, one group of participants may be trained to value effort using one of the manipula-tions introduced by Clay et al. [62] and Lin et al. [63]. If our results were truly driven by altered representations of effort costs, then this group should also show a similar attenuation of the relief effect.

Finally, in Experiment 2 we also found that participants valued positive feedback after the arithmetic task more highly when they had successfully performed the hard compared to easy task. This result suggests that effort expenditure (as opposed to anticipation only) can also influence reward processing (see also [66, 67]), retrospectively increasing the value of rewards when more effort is exerted. This finding bears resemblance to the IKEA effect [58, 68], according to which people value a product more if it is produced by their own efforts com-pared to the same product acquired by other means. In addition, it calls to mind the idea that self-efficacy [69], typically achieved through effort, carries value, and contributes to a positive sense of competence (c.f. self-determination theory [70]).

Some methodological limitations warrant a final note. The current studies used a relatively small sample of student participants. This is especially noteworthy for Experiment 2, where we aimed to assess associations between individual differences. Our results invite a replication with a larger and more heterogeneous sample. Second, our effects emerged in the context of a simple gambling task, and so it is unknown whether the relief effect would emerge in other contexts. For example, what would happen if reward feedback, unlike the current study, would be useful for decision making (e.g., in the context of reinforcement learning)? Third, we estab-lished a robust relief effect based only on subjective ratings of the reward-related feedback, but it is unclear whether this novel paradigm would still elicit its neural correlates [1]. Neuroimag-ing work using electrophysiology would not only be useful for validating the relief effect at the single trial level, but also for developing formal neurocomputational models of cost-benefit tra-deoffs in effort allocation.

In sum, this study provides novel evidence for the notion that cognitive effort carries a cost, and that this cost is tightly linked to both reward processing and cost-benefit analyses of effort allocation. More precisely, our results suggest that if anticipated effort is avoided, it is returned as a reward, and experienced as relief. Intriguingly, this effect is attenuated in people who report being more disposed towards effortful tasks, suggesting they are less sensitive to cogni-tive costs. This raises the outstanding question of whether differences in NFC mitigate the aversive nature of cognitive effort, or whether they result in heightened stakes of engaging with, and succeeding at, hard cognitive tasks.

## Supporting information

**S1 File.**
(PDF)

## Author Contributions

**Conceptualization:** Davide Gheza, Wouter Kool, Gilles Pourtois.

**Data curation:** Davide Gheza.

**Formal analysis:** Davide Gheza.

**Investigation:** Davide Gheza, Gilles Pourtois.

**Methodology:** Davide Gheza.

**Project administration:** Davide Gheza.

**Resources:** Gilles Pourtois.

**Software:** Davide Gheza.

**Supervision:** Davide Gheza, Gilles Pourtois.

**Validation:** Davide Gheza.

**Visualization:** Davide Gheza.

**Writing – original draft:** Davide Gheza, Wouter Kool, Gilles Pourtois.

**Writing – review & editing:** Davide Gheza, Wouter Kool, Gilles Pourtois.

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
