## [Decision Letter · Decision Letter 0]

31 Jul 2023

PONE-D-23-18706Need for cognition moderates cognitive effort aversionPLOS ONE

Dear Dr. Gheza,

Thank you for submitting your manuscript to PLOS ONE. Two reviewers have read your manuscript and as you can see below they both enjoyed reading your study but had a number of questions regarding analyses and interpretations that will need your attention. Thus, we feel that your manuscript has merit but does not fully meet PLOS ONE’s publication criteria as it currently stands. We invite you to submit a revised version of the manuscript that addresses the points raised during the review process.

We look forward to receiving your revised manuscript.

Kind regards,

Poppy Watson

Academic Editor

PLOS ONE

Journal Requirements:

Reviewers' comments:

Reviewer's Responses to Questions

**Comments to the Author**

1. Is the manuscript technically sound, and do the data support the conclusions?

Reviewer #1: Partly

Reviewer #2: Yes

2. Has the statistical analysis been performed appropriately and rigorously? 

Reviewer #1: Yes

Reviewer #2: Yes

3. Have the authors made all data underlying the findings in their manuscript fully available?

Reviewer #1: Yes

Reviewer #2: Yes

4. Is the manuscript presented in an intelligible fashion and written in standard English?

Reviewer #1: Yes

Reviewer #2: Yes

5. Review Comments to the Author

Reviewer #1: Overview: This paper investigates whether avoiding anticipated cognitive effort is itself rewarding. The paper shows, via two experiments using a gambling paradigm, that peoples' sense of relief when avoiding effort is dependent on how much effort an individual was anticipating. Relatedly, the authors argue this sense of relief is moderated by a persons’ Need for Cognition.

Overall, the paper is clearly written and I believe the research question is interesting. The results are well presented and generally align with the authors’ hypotheses—people do seem to be more relieved when avoiding high effort tasks as opposed to low effort tasks. The analysis approach is also appropriate (although could be improved) and the figures present the data transparently. I do however have some reservations about the strengths of the conclusions drawn, plus some other questions and suggestions.

1. Accuracy and self-report scores. It is possible that differences in subjective ratings (pleasantness, frustration, relief) arose due to differences in participants’ expectations of reward, rather than as a direct consequence of avoiding effort. Consider this rough calculation using the accuracy data from Exp 1: on a ‘hard’ trial, the average participant has a 50% chance of immediate reward, a 43.5% chance of reward following effort (87% avg accuracy) and a 6.5% chance of no reward; on an ‘easy’ trial a participant has a 50% chance of immediate reward, a 49% chance of reward following effort (98% avg. accuracy), and 1% no reward. The EVs for the two trial types, hard and easy, are therefore 5.61 points and 5.94, respectively. A participant may therefore feel more relieved on hard trials as it was less likely, as a function of how hard the arithmetic task was, they would receive reward. This hypothesis is somewhat supported by your finding that participants are more relieved after getting ‘hard’ trials correct than ‘easy’ trials (Figure 7). One solution would be to control for participants’ accuracy in the experiment. If a model containing ‘difficulty level’ accounted for the data better than a model containing participants’ accuracy (all other model components being equal), it would provide stronger evidence that effort avoidance is driving the ‘relief effect’. Note that Westbrook et al. (2013) use such an approach to assess effort discounting.

2. Analysis methods. I’ve assumed, as it isn’t entirely clear from the paper, that you are currently averaging participants’ responses and then using participant averages to fit your models. Given each participant gives repeated estimates, it may be more appropriate to use a Bayesian mixed modelling approach which eliminates the need for such aggregation. This can be done vis the brms package in R and would allow the assessment of both fixed and random effects.

3. Relationship between NFC and accuracy. I’m glad the authors included this analysis. I do however think the authors could introduce some scepticism as to the causal direction of this relationship. Given participants fill out the NFC scale immediately after the gambling task is completed, it is possible their NFC scores are affected by their accuracy, rather than NFC affecting accuracy. I also think it would be worthwhile including accuracy as a factor in the analysis which assesses the moderating role of NFC on people’s self-report ratings (pleasantness, frustration, relief). At the very least this potential limitation/confound should be discussed.

4. I wondered why you chose to aggregate across the different subjective rating types (i.e., pleasantness, frustration, relief) given your primary interest, as far as the introduction is concerned, is in a sense of relief. This effect is explicitly referred to as the ‘relief effect’ but comprises frustration and pleasantness scores in addition to relief. Naturally, you have more statistical power by combining these ratings, but it seems like an odd choice to deliberately include 3 different rating types and then aggregate across them in your analysis. I think the authors could include a clearer rationale for their current approach.

Minor points

• In lines 431-432 the authors argue the results show people exerted more effort in the hard condition. This doesn’t seem obvious to me. Perhaps participants exerted the same amount of effort (which itself is difficult to define) and that is why they performed worse at the task.

• What was the scale/range of the VAS ratings? They have been converted to percentages for analysis, but what was the original scale? This is not reported.

• I don’t think Figure 4A adds much and the authors could just report Figure 4B. I’m also not sure if it is necessary to include Figure 5 as it doesn’t add much in addition to Figure 4, but, if you are going to keep it you should explain how the participants in the two extremes are categorised (i.e., bottom and top 30%) in the main text as opposed to only in the Figure caption.

• I don’t think the current title does a great job of representing the work. The fact that NFC is related to effort aversion is well established. I think you could focus on the relief aspect, which is framed as your primary interest in the abstract.

• At a few points the authors alternate between using a period (.) and a comma (,) as a decimal separator (e.g., line 195). I imagine the norm for this journal would be a period.

• Consistency with the number of decimal places would also be good. For example, the accuracy scores are reported to the nearest integer, but subjective ratings are to 1 decimal place. Reporting to two decimal places throughout makes sense to me.

• Line 459-60 refers to the “law of least mental effort”, given Hull was not specifically referring to cognitive effort in his work it may be better to drop the “mental” from this phrase.

Jake Embrey

I sign all of my reviews

Reviewer #2: The current study aimed to test previously published findings from the same group about a relief effect that occurs when participants avoided having to spend mental effort in a gambling task. Across two experiments, they found that participants reported more pleasure, less frustration, and more relief for outcomes that signaled successful avoidance of effort, especially if the anticipated effort was high compared to low. This effect was reversed for outcomes signaling that effort had to be exerted. In the second experiment, the authors show that NFC scores moderate the strength of these results.

Overall, I think the manuscript is well-written and that this study tackles a relevant topic that will be of interest to the broader readership of the journal. The rationale is well-explained, and the design is well-suited to answer the research question. Although I think that the study deserves to be published, I have a few comments that should be addressed before I can ultimately recommend the manuscript to be accepted. These comments are mostly related to the statistical analyses and the authors’ interpretation of the data.

Major comments:

- It was a little unclear to me what exactly served as the dependent variable of interest here. Plots depict participants’ ratings for pleasure, frustration, and relief separately, but it seems as if the authors actually used an average score across these dimensions in their ANOVAs. Is this correct? If not, please ignore the following points. If it is, it should be explained more specifically. The resulting effect of these ANOVAs is sometimes referred to as a relief effect, but if all outcome-related ratings are averaged, it is not really (only) about relief but more generally about the affective response (and sometimes it is actually referred to as such). The authors also seem to interpret this as a measure of effort aversion (hence the title of the manuscript). So, my point here is, please explain how you arrived at your DV and use a consistent term to refer to this effect throughout the manuscript. I suggest it should not be the term relief, though, as it includes more than the specific feeling of relief.

- Related to the previous point, what was the reason to combine these ratings instead of looking at them separately?

- A strength of the paper is that the authors use Bayes Factor comparisons to find the most appropriate models for their analyses. The corresponding numbers are in the text, but it might help the reader to put them in a table (similar to the guidelines published here: https://osf.io/wae57). This applies to both experiments. In experiment 2, a table for the output of the brms model might be good, too.

- I was also unclear about the random effects specification of the models. On page 11, the authors state that they included participant, affective dimension, and their interaction in their models. How exactly did this look like? Would it have made sense to include affective dimension as a fixed effect and nest everything within participants instead, so that potential differences between these dimensions could have been analyzed? Here, again, I am mostly arguing for more clarity and justification (instead of doubting/criticizing the authors’ decisions).

Minor comments:

- Title: I was wondering if the title of the manuscript (that really emphasizes the moderating effect of NFC) truly captures the main findings of the study well, given that NFC is only measured in the second experiment and that the main purpose across both experiments was more to validate positive affective responses associated with not having to exert effort. To be clear, I wouldn’t object the current title if the authors choose to stick with it, I just think there might be better alternatives.

- On page 20, lines 485 – 487, the authors mention that NFC was measured after the task and that the questionnaire was thus unlikely to influence performance in the task itself. This is reasonable. However, I was wondering whether having experienced the task and the subjective responses to potential or actual effort exertion may have affected how participants responded to these questions (i.e., if they did not enjoy completing the math trials, they might be inclined to rate their overall motivation to engage in mentally demanding behavior lower in the questionnaire). I know NFC is supposed to be a trait, but the task might still have some transient effect on participants’ ratings that might explain the relationship between ratings and behavior. This is in no way a major problem for the study, but it might be worth a sentence or two in the discussion.

- This is more a question out of curiosity than a comment: Figures 2 and 3 show that in both experiments, there were people who rated relief low in reward feedback trials and high in no-reward feedback trials. This is a little surprising, although it is unclear from the plots whether these participants were at least internally consistent (i.e., a person who rated relief as let’s say 25 for hard trials with reward feedback would rate it even lower for hard trials with no-reward feedback). Did participants show this consistency or were there some who displayed ratings that ran completely against expectations (i.e., more relief, pleasure etc. for no-reward trials than for reward trials)?

6. PLOS authors have the option to publish the peer review history of their article (what does this mean?). If published, this will include your full peer review and any attached files.

Reviewer #1: **Yes: **Jake R. Embrey

Reviewer #2: No

---

## [Author Response · Author response to Decision Letter 0]

30 Sep 2023

Please refer to the Response to Reviewer pdf file for a better formatting and readability.

PONE-D-23-18706

Need for cognition moderates cognitive effort aversion PLOS ONE

Reviewer 1

 Overview: This paper investigates whether avoiding anticipated cognitive effort is itself rewarding. The paper shows, via two experiments using a gambling paradigm, that peoples' sense of relief when avoiding effort is dependent on how

 much effort an individual was anticipating. Relatedly, the authors argue this sense

 of relief is moderated by a persons’ Need for Cognition.

 Overall, the paper is clearly written and I believe the research question is interesting. The results are well presented and generally align with the authors’ hypotheses—people do seem to be more relieved when avoiding high effort tasks as opposed to low effort tasks. The analysis approach is also appropriate (although could be improved) and the figures present the data transparently. I do however have some reservations about the strengths of the conclusions drawn, plus some

 other questions and suggestions.

We thank the reviewer for the kind words, and for the thoughtful points. We believe that they have allowed us to improve the paper.

1. Accuracy and self-report scores. It is possible that differences in subjective ratings (pleasantness, frustration, relief) arose due to differences in participants’ expectations of reward, rather than as a direct consequence of avoiding effort. Consider this rough calculation using the accuracy data from Exp 1: on a ‘hard’ trial, the average participant has a 50% chance of immediate reward, a 43.5% chance of reward following effort (87% avg accuracy) and a 6.5% chance of no reward; on an ‘easy’ trial a participant has a 50% chance of immediate reward, a 49% chance of reward following effort (98% avg. accuracy), and 1% no reward. The EVs for the two trial types, hard and easy, are therefore 5.61 points and 5.94, respectively. A participant may therefore feel more relieved on hard trials as it was less likely, as a function of how hard the arithmetic task was, they would receive reward. This hypothesis is somewhat supported by your finding that participants are more relieved after getting ‘hard’ trials correct than ‘easy’ trials (Figure 7). One solution would be to control for participants’ accuracy in the experiment. If a model containing ‘difficulty level’ accounted for the data better than a model containing participants’ accuracy (all other model components being equal), it would provide stronger evidence that effort avoidance is driving the ‘relief effect’. Note that Westbrook et al. (2013) use such an approach to assess effort discounting.

We thank the reviewer for bring this concern to our attention. They correctly point out that the difference in accuracy between effort conditions results in a difference in reward probability (albeit a relatively small difference, as well quantified by the reviewer). Therefore, we have run control analyses that control for experienced difficulty when

predicting the feedback ratings. More precisely, we computed the average accuracy scores for each difficulty level and participant, and we use these as a continuous predictor (instead of the factor difficulty level used before), as a proxy of expected values for each condition. Following the reviewer’s logic, we reasoned that if the model including ‘difficulty level’ outperformed the model including ‘accuracy’, it would be unlikely that reward expectations (rather than expected effort) drove our effect. One caveat to this reasoning is that individual difference in performance may as well act as a more sensitive proxy of expected effort for the two difficulty conditions, because it models variance in the relative difference in difficulty between conditions across participants. Therefore, this control analysis should not be interpreted as decisive in ultimately disentangling these two alternative interpretations.

Nevertheless, we compared two models including either the outcome x difficulty or outcome x accuracy for both experiments. In Experiment 1, we found that data where more likely under a model including outcome x accuracy than under a model including outcome x difficulty. In Experiment 2, we found the opposite result. The data were more likely under a model including outcome x difficulty than under a model including outcome x accuracy. We report these results in the Supporting Information.

In short, it is possible that the effect of difficulty level affected the feedback ratings through changes in expected value in Experiment 1. On the other hand, we find evidence against this hypothesis for Experiment 2. Importantly, the most important claims of our paper are based on this second experiment. We conclude that our experimental design is not suited to fully address this potential confound. Future research may address this issue with different experimental procedures, for example by omitting performance feedback. We discuss these points in a new paragraph in the General Discussion, where we also reference the analyses in the Supplemental Materials:

"Another possibility is that participants scoring low in NFC, and performing sub-par in the arithmetic task, may have developed lower expected values for high difficulty trials, corresponding to lower estimates of reward probability. Therefore, the relief for avoiding high effort may be compounded by the relief for avoiding risky arithmetic trials. In the Supporting Information we report two control analyses which confirm the relief effect, and the role of NFC in moderating it, even when controlling for individual differences in arithmetic performance.

However, because individual differences in performance may index relative differences in expected effort between conditions, our experimental design is not suited to ultimately adjudicate between these competing, but not mutually exclusive, accounts. Future research may address this issue with different experimental procedures, for example by omitting performance feedback.”

Please note that this text refers to the results of the control analysis reported above (comparing the use of difficulty as factor vs. accuracy as a continuous predictor), but also to another control analysis suggested by the reviewer, which we discuss in detail in point 3 below (where we use the difference in average accuracy between difficulty levels as additional or alternative moderator to NFC).

2. Analysis methods. I’ve assumed, as it isn’t entirely clear from the paper, that you are currently averaging participants’ responses and then using participant averages to fit your models. Given each participant gives repeated estimates, it may be more appropriate to use a Bayesian mixed modelling approach which eliminates the need for such aggregation. This can be done vis the brms package in R and would allow the assessment of both fixed and random effects.

We thank the reviewer for bringing this concern to our attention. It has allowed us to clarify our paper. In fact, our analytic approach already fully relies on (hierarchical) mixed-effect models. In all main analyses, we model single-trial data using hierarchical regression with subjects and affective dimensions as random effects. Because we strived to keep these main analyses simple and accessible, we opted for the Bayes Factor approach. However, when we turned our attention to the more complex 3-way interaction (including outcome x difficulty level x NFC; cf. model 5), we corroborated the main analyses with full Bayesian hierarchical regressions using the BRMS package (like the reviewer suggested). We have added text to the Statistical Analyses section to highlight this. On page 10, we now say:

“Subjective ratings for the three affective dimensions were used to assess the hedonic response as a whole, along a negative-positive valence axis (41). In all analyses, we predicted single trial subjective ratings with a mixed modelling approach, in which participant and affective dimension were treated as random effects (i.e., considered nuisance factors).”

On page 12, we add:

“In Experiment 2, to further explore the effect of NFC on reward processing, we also ran a full Bayesian estimation of parameter values for model 5, using the “brms” R package.”

3. Relationship between NFC and accuracy. I’m glad the authors included this analysis. I do however think the authors could introduce some scepticism as to the causal direction of this relationship. Given participants fill out the NFC scale immediately after the gambling task is completed, it is possible their NFC scores are affected by their accuracy, rather than NFC affecting accuracy. I also think it would be worthwhile including accuracy as a factor in the analysis which assesses the moderating role of NFC on people’s self-report ratings (pleasantness, frustration, relief). At the very least this potential limitation/confound should be discussed.

We agree with the reviewer, and have added the following text to the General Discussion (page 21):

“Rather, it is possible that experiencing the task may have biased participants’ responses in the NFC questionnaire. Perhaps, observing their performance on the task influenced the self-evaluations prompted by the NFC questionnaire, leading their scores to reflect a combination of stable trait features with transient affective reactions”.

We added this sentence to a paragraph in which we already placed some caution about interpreting changes in the relief effect as purely driven by difference in the motivation to engage with effort (since we found a relationship between NFC and arithmetic performance).

We also ran the new analyses suggested by the reviewer, and we report them in the Supporting Information. Specifically, we ran an additional control analysis including accuracy when assessing the moderating role of NFC on people’s self-report ratings. We defined a new continuous predictor, “delta_performance”, corresponding to the difference in mean accuracy in the hard vs. easy conditions. Next, we added this predictor in a new model “outcome*effort*delta_accuracy”. We reasoned that the larger the delta_performance, the larger the difference in expected reward between conditions. We expected this interaction to compete for some of the variance explained by the outcome*effort*NFC, but we predicted that a model including the interactions with delta_accuracy (outcome*effort*delta_accuracy) and the interactions with NFC (outcome*effort*NFC) would explain the data better than one including the interactions with delta_accuracy alone. Results confirmed this prediction. Moreover, even a simple three-way interaction model including NFC performed much better than the three-way interaction model including delta_accuracy.

As mentioned above, we briefly describe this analysis in the General Discussion and refer to the Supporting Information there.

4. I wondered why you chose to aggregate across the different subjective rating types (i.e., pleasantness, frustration, relief) given your primary interest, as far as the introduction is concerned, is in a sense of relief. This effect is explicitly referred to as the ‘relief effect’ but comprises frustration and pleasantness scores in addition to relief. Naturally, you have more statistical power by combining these ratings, but it seems like an odd choice to deliberately include 3 different rating types and then aggregate across them in your analysis. I think the authors could include a clearer rationale for their current approach.

We thank the reviewer for pointing out this ambiguity, and for giving us the opportunity to better clarify our rationale. Our goal was to assess a broad hedonic response towards the anticipation or likely avoidance of effort. Therefore, we used three affective dimensions, with different semantic connotations, so that we could collect converging evidence along a single axis (positive to negative). In addition, this approach allowed us to validate the consistency of participants’ reports. We now justify this choice more explicitly in the Stimuli and task section (page 8):

“By using three questions, which assessed a range of different affective dimensions, we aimed at measuring the latent construct of a ‘hedonic response’ along a negative-positive valence axis (41). Multiple self-report items allowed us to validate the consistency of participants’ responses, and to increase statistical power by controlling for nuisance variability between affective dimensions.”

We feel this choice is especially warranted because the measures show extremely robust correlations. The average correlation for frustration and pleasure was 0.81 (Exp. 1) and 0.67 (Exp. 2). The average correlation for frustration and relief was 0.71 (Exp. 1) and 0.57 (Exp. 2). The average correlation for pleasure and relief was 0.85 (Exp. 1) and 0.80 (Exp. 2). We report these correlations in the Supporting Information, and refer to them in the text on page 14:

“Moreover, we observed very robust correlations between pairs of affective dimensions (group-level statistics of these correlations are reported in Supporting Information).”

We also clarify the distinction between the latent construct that we wanted to measure (hedonic responses) and the interpretation of its modulation by the factors Outcome and Difficulty level (the ‘relief effect’). In the Introduction (page 5), we now say:

“In line with our previous study, we assessed the latent construct of the “hedonic response” (i.e., the subjective experience of the gambling trial outcome) using three probes, each addressing a different affective dimension (“pleasure”, “frustration”, and “relief”). We refer to the interaction of trial outcome with effort anticipation on the hedonic response as a “relief effect”.”

We also added a clarification on the Statistical analysis section (pages 11-12):

“Moreover, the factors Subject, Affective dimension, and their interaction were treated as nuisance variables (i.e., varying effects to control for). To do this, they were entered in all models, including the null model. Additionally, the null model included a single (i.e., constant) intercept. This procedure allowed us to control for the nuisance effects of 1) individual differences in VAS biases (e.g., preference for a given section of the scale), 2) overall differences in rating across affective dimensions (e.g., overall higher responses for a specific scale), and 3) individual variations in the latter differences.”

Finally, we report the results of analyses when only modeling the relief component in the Supporting Information. These results were qualitatively and quantitively similar to the results reported in the main paper. We have included a footnote in the main text that refers the reader to this analysis:

“When selectively predicting the feedback ratings for the ‘relief’ affective dimension, data were still best explained by a model that included the three-way interaction of outcome x difficulty level x NFC, followed by a model including the two-way outcome x difficulty level interaction (see Supporting Information).”

Minor points

- In lines 431-432 the authors argue the results show people exerted more effort in the hard condition. This doesn’t seem obvious to me. Perhaps participants exerted the same amount of effort (which itself is difficult to define) and that is why they performed worse at the task.

This is a valid point worth of further discussion. To clarify, we believe that our results are only suggestive of such an interpretation (i.e., that “people allocated more effort in the hard condition”). We base this claim on the relatively minor difference in accuracy, in the face of substantial difference in difficulty, and (importantly) a lack of clear change in reported motivation. In other words, we think that the overall pattern of results lends some support for the conclusion that participants exerted more effort in the hard condition, but this conclusion is not fully granted. We now clarify the caveat of this interpretation, by adapting the concluding sentence of that paragraph:

“We believe that our data shows that participants exerted more cognitive effort in the hard condition. In response to a substantial difference in difficulty, they only showed a relatively minor drop in accuracy. Most importantly, we saw no clear shift in reported motivation. Nevertheless, it is important to note that our findings provide only preliminary support for this conclusion, as both task demands and the precision of the information-processing operations (i.e., performance) differed, to some extent, between difficulty conditions (13).”

- What was the scale/range of the VAS ratings? They have been converted to percentages for analysis, but what was the original scale? This is not reported.

The visual analog scale (VAS) was presented as a continuous line segment at the center of the screen, bounded with vertical shorter segments. It ranged 396 pixels of the screen. To compute the VAS percentage, we converted the mouse coordinates on the screen along the VAS line. We first calculated the relative pixel position of the mouse at time of click, with respect to the leftmost point of the scale. We then converted this point in percentage, considering the 396 pixels range. We have added a sentence in the Data analysis section (page 10) to explain this in more detail:

“VAS scores were calculated by first identifying the relative x-axis position of the mouse with respect to the leftmost position of the scale at the time of participant’s click. We then converted this measure from pixels distance to percentage over the scale range (396 pixels).”

- I don’t think Figure 4A adds much and the authors could just report Figure 4B. I’m also not sure if it is necessary to include Figure 5 as it doesn’t add much in addition to Figure 4, but, if you are going to keep it you should explain how the participants in the two extremes are categorised (i.e., bottom and top 30%) in the main text as opposed to only in the Figure caption.

We agreed with the reviewer that Figure 4A was redundant, and have removed it. However, we have decided to keep Figure 5, insofar it provides different information (i.e., raw data) compared to Figure 4 (i.e., estimates based on posterior distributions). Of course, if the reviewer feels strongly about this, we’d be happy to consider removing it.

We now also explain in text how the extremes are categorized (page 17):

“(participants within the 0-30 percentiles were included in the Low NFC group; participants within the 70-100 percentiles were included in the High NFC group)”

- I don’t think the current title does a great job of representing the work. The fact that NFC is related to effort aversion is well established. I think you could focus on the relief aspect, which is framed as your primary interest in the abstract.

We agree with the reviewer and have changed the title to “Need for cognition moderates the relief of avoiding cognitive effort”.

- At a few points the authors alternate between using a period (.) and a comma (,) as a decimal separator (e.g., line 195). I imagine the norm for this journal would be a period.

We thank the reviewer for spotting these typos. We now consistently use periods as the decimal separator.

- Consistency with the number of decimal places would also be good. For example, the accuracy scores are reported to the nearest integer, but subjective ratings are to 1 decimal place. Reporting to two decimal places throughout makes sense to me.

We have made these changes.

- Line 459-60 refers to the “law of least mental effort”, given Hull was not specifically referring to cognitive effort in his work it may be better to drop the “mental” from this phrase.

Agreed.

Reviewer 2

The current study aimed to test previously published findings from the same group about a relief effect that occurs when participants avoided having to spend mental effort in a gambling task. Across two experiments, they found that participants reported more pleasure, less frustration, and more relief for outcomes that signaled successful avoidance of effort, especially if the anticipated effort was high compared to low. This effect was reversed for outcomes signaling that effort had to be exerted. In the second experiment, the authors show that NFC scores moderate the strength of these results.

Overall, I think the manuscript is well-written and that this study tackles a relevant topic that will be of interest to the broader readership of the journal. The rationale is well-explained, and the design is well-suited to answer the research question. Although I think that the study deserves to be published, I have a few comments that should be addressed before I can ultimately recommend the manuscript to be

accepted. These comments are mostly related to the statistical analyses and the authors’ interpretation of the data.

We thank the reviewer for their positive remarks, and also for their more critical comments. We believe these comments strengthened the manuscript.

1. It was a little unclear to me what exactly served as the dependent variable of interest here. Plots depict participants’ ratings for pleasure, frustration, and relief separately, but it seems as if the authors actually used an average score across these dimensions in their ANOVAs. Is this correct? If not, please ignore the following points. If it is, it should be explained more specifically. The resulting effect of these ANOVAs is sometimes referred to as a relief effect, but if all outcome-related ratings are averaged, it is not really (only) about relief but more generally about the affective response (and sometimes it is actually referred to as such). The authors also seem to interpret this as a measure of effort aversion (hence the title of the manuscript). So, my point here is, please explain how you arrived at your DV and use a consistent term to refer to this effect throughout the manuscript. I suggest it should not be the term relief, though, as it includes more than the specific feeling of relief.

We thank the reviewer for pointing out this ambiguity, and for giving us the opportunity to clarify. We should note that Reviewer 1 raised a very similar point (see above). Therefore, we address them together.

First, we should clarify that we do not average scores across the three affective dimensions, but rather we predict each individual rating, separately for each affective dimension. As explained below, we control for differences across dimensions by adding the factor Affective dimension to all our models, including the null.

This said, it is true that we aim at assessing the latent construct of “hedonic response” as a whole. More specifically, we used three affective dimensions, with different semantic connotations, so that we could collect converging evidence along a single axis (positive to negative). In addition, this approach allowed us to validate the consistency of participants’ reports. We now justify this choice more explicitly in the Stimuli and task section (page 8):

“By using three questions, which assessed a range of different affective dimensions, we aimed at measuring the latent construct of a ‘hedonic response’ along a negative-positive valence axis (41). Multiple self-report items allowed us to validate the consistency of participants’ responses, and to increase statistical power by controlling for nuisance variability between affective dimensions.”

We feel this choice is especially warranted because the measures show extremely robust correlations. The average correlation for frustration and pleasure was 0.81 (Exp. 1) and 0.67 (Exp. 2). The average correlation for frustration and relief was 0.71 (Exp. 1) and 0.57 (Exp. 2). The average correlation for pleasure and relief was 0.85 (Exp. 1) and 0.80 (Exp.

2). We report these correlations in the Supporting Information, and refer to them in the text on page 14:

“Moreover, we observed very robust correlations between pairs of affective dimensions (group-level statistics of these correlations are reported in the Supporting Information).”

We also clarify the distinction between the latent construct that we wanted to measure (hedonic responses) and the interpretation of its modulation by the factors Outcome and Difficulty level (the ‘relief effect’). In the Introduction (page 5), we now say:

“In line with our previous study, we assessed the latent construct of the “hedonic response” (i.e., the subjective experience of the gambling trial outcome) using three probes, each addressing a different affective dimension (“pleasure”, “frustration”, and “relief”). We refer to the interaction of trial outcome with effort anticipation on the hedonic response as a “relief effect”.”

We also added a clarification on the Statistical analysis section (pages 11-12):

“Moreover, the factors Subject, Affective dimension, and their interaction were treated as nuisance variables (i.e., varying effects to control for). To do this, they were entered in all models, including the null model. Additionally, the null model included a single (i.e., constant) intercept. This procedure allowed us to control for the nuisance effects of 1) individual differences in VAS biases (e.g., preference for a given section of the scale), 2) overall differences in rating across affective dimensions (e.g., overall higher responses for a specific scale), and 3) individual variations in the latter differences.”

In some of the Figure captions (Fig. 2,3,7) we now clarify that we combine the three affective dimensions in the statistical analysis:

“Reponses to the three affective dimensions were used to estimate the underlying hedonic response along a single valence axis.”

Finally, we report the results of analyses when only modeling the relief component in the Supporting Information. These results were qualitatively and quantitively similar to the results reported in the main paper. We have included a footnote in the main text that refers the reader to this analysis:

With these changes, we hope to have better clarified the distinction between our dependent variable and the latent construct we aimed to measure. To emphasize this, we now use the term “hedonic response” more consistently across the manuscript.

2. Related to the previous point, what was the reason to combine these ratings instead of looking at them separately?

We believe we have already addressed this question in our previous answer, and we elaborate on it below (point 4). In short, we leveraged the statistical strength of mixed models to control for consistency across responses.

3. A strength of the paper is that the authors use Bayes Factor comparisons to find the most appropriate models for their analyses. The corresponding numbers are in the text, but it might help the reader to put them in a table (similar to the guidelines published here: https://osf.io/wae57). This applies to both experiments. In experiment 2, a table for the output of the brms model might be good, too.

We have added two tables reporting the model comparison results for each experiment, and a table for the parameter estimates from “brms” for Experiment 2.

4. I was also unclear about the random effects specification of the models. On page 11, the authors state that they included participant, affective dimension, and their interaction in their models. How exactly did this look like? Would it have made sense to include affective dimension as a fixed effect and nest everything within participants instead, so that potential differences between these dimensions could have been analyzed? Here, again, I am mostly arguing for more clarity and justification (instead of doubting/criticizing the authors’ decisions).

We thank the reviewer for giving us the opportunity to better explain how we specified the nuisance effects (i.e., the varying effects, also referred to as ‘random’). We have made substantial changes to Statistical Analysis paragraph referenced by the reviewer. On pages 11-12, we say:

“Moreover, the factors Subject, Affective dimension, and their interaction were treated as nuisance variables (i.e., varying effects to control for). To do this, they were entered in all models, including the null model. Additionally, the null model included a single (i.e., constant) intercept. This procedure allowed us to control for the nuisance effects of 1) individual differences in VAS biases (e.g., preference for a given section of the scale), 2) overall differences in rating across affective dimensions (e.g., overall higher responses for a specific scale), and 3) individual variations in the latter differences.”

It was our deliberate intention to treat that Affective dimension factor as nuisance, since we did not have sufficient theoretical ground to make predictions about potential differences between its levels. Rather, as explained above, we used three different Affective dimension levels with the aim of assessing the broad hedonic response of the participants along a ‘valence’ axis (from negative to positive). First, probing the same latent construct (hedonic response) with three different questions allowed us to increase the statistical power of our analysis. Second, the three dimensions allowed us to ensure the consistency of participant’s reports. Specifically, in case of inconsistent responses across dimensions, their variance would have been explained by the nuisance effects, and the participant’s contribution to the constant effects of interest would have been hindered.

We also noticed a mistake in the statistical analyses of Experiment 2 (R script for Exp 2 only). In particular, the random term “Subject” in some of the models was treated as a numeric variable, rather than a factor. We have corrected this error and the text reflects the new results (pp. 15-16). Importantly, this correction does not alter the pattern of results described before, but rather provides even stronger evidence in the same direction.

Minor comments

- Title: I was wondering if the title of the manuscript (that really emphasizes the moderating effect of NFC) truly captures the main findings of the study well, given that NFC is only measured in the second experiment and that the main purpose across both experiments was more to validate positive affective responses associated with not having to exert effort. To be clear, I wouldn’t object the current title if the authors choose to stick with it, I just think there might be better alternatives.

We agree with the reviewer and have changed the title to “Need for cognition moderates the relief of avoiding cognitive effort”.

- On page 20, lines 485 – 487, the authors mention that NFC was measured after the task and that the questionnaire was thus unlikely to influence performance in the task itself. This is reasonable. However, I was wondering whether having experienced the task and the subjective responses to potential or actual effort exertion may have affected how participants responded to these questions (i.e., if they did not enjoy completing the math trials, they might be inclined to rate their overall motivation to engage in mentally demanding behavior lower in the questionnaire). I know NFC is supposed to be a trait, but the task might still have some transient effect on participants’ ratings that might explain the relationship between ratings and behavior. This is in no way a major problem for the study, but it might be worth a sentence or two in the discussion.

We thank the reviewer for suggesting this subtle but relevant element. We agree that experiencing the task, which includes mental arithmetic, may have induced a transient affective reaction. Responses to the NFC questionnaire, albeit putatively assessing a stable trait, may have been influenced by these affective aftereffects of the task. We included this consideration in the discussion, on page 23.

“Rather, it is possible that experiencing the task may have biased participants’ responses in the NFC questionnaire. Perhaps, observing their performance on the task influenced the self-evaluations prompted by the NFC questionnaire, leading their scores to reflect a combination of stable trait features with transient affective reactions”.

- This is more a question out of curiosity than a comment: Figures 2 and 3 show that in both experiments, there were people who rated relief low in reward feedback trials and high in no-reward feedback trials. This is a little surprising, although it is unclear from the plots whether these participants were at least internally consistent (i.e., a person who rated relief as let’s say 25 for hard trials with reward feedback would rate

it even lower for hard trials with no-reward feedback). Did participants show this consistency or were there some who displayed ratings that ran completely against expectations (i.e., more relief, pleasure etc. for no-reward trials than for reward trials)?

To address this point, we have added Supplementary Figures 1-6 in the Supporting Information:

 100 75 50 25 0

Com

150 positera

100 ting(%

)

50

reward−noreward conditions

) %(g nitaR

Easy Easy reward no−reward

Hard Hard reward no−reward

Relief

e.g., Supplementary Figure 3 (Experiment 1)

They recapitulate the results in the main Figs 2-3, but each Figure refers to a single affective dimension, and compares reward to no-reward conditions for each difficulty level. Moreover, we connected participant’s datapoints with colored segments to highlight the pattern of ratings across conditions. Last, the secondary plot on the right shows a composite rating, obtained by summing ratings from reward conditions and subtracting the ones from no-reward conditions. As can be seen from the figures, there is overall internal consistency across reward levels (i.e., lower affective value for no-reward vs reward). This is especially the case for the affective dimensions “relief” and “pleasure”, while a few participants showed a modest reversal of this pattern for the dimension “frustration”, albeit in Experiment 2 only (negative values in the composite rating). Some participants showed less variance than others in their responses across different conditions, demonstrating a preference for a narrow section of the VAS scale (i.e., showing high precision in their ratings across conditions). This translates into a flatter pattern across conditions. Nevertheless, the data shows a robust internal consistency. We added a reference to these supplementary results in the Result section (page 14):

“As can be seen from the Supplementary Figures 1-6, participants showed a robust internal consistency in their rating of the reward-related feedback across outcome levels: Their rating was consistently lower for no-reward compared to reward outcome.”

---

## [Decision Letter · Decision Letter 1]

12 Oct 2023

PONE-D-23-18706R1Need for cognition moderates the relief of avoiding cognitive effortPLOS ONE

Dear Dr. Gheza,

Thank you for resubmitting your manuscript to PLOS ONE. As you can see, both reviewers feel that the manuscript is vastly improved and is nearly ready for publication. There is one final point which reviewer 1 has raised and it would be great if you could consider this additional analysis for inclusion.  I do not intend to send the manuscript back out for re-review, provided that you can clearly address the reviewer's comment in your response letter/manuscript.

We look forward to receiving your revised manuscript.

Kind regards,

Poppy Watson

Academic Editor

PLOS ONE

Journal Requirements:

Reviewers' comments:

Reviewer's Responses to Questions

**Comments to the Author**

1. If the authors have adequately addressed your comments raised in a previous round of review and you feel that this manuscript is now acceptable for publication, you may indicate that here to bypass the “Comments to the Author” section, enter your conflict of interest statement in the “Confidential to Editor” section, and submit your "Accept" recommendation.

Reviewer #1: (No Response)

Reviewer #2: All comments have been addressed

2. Is the manuscript technically sound, and do the data support the conclusions?

Reviewer #1: Yes

Reviewer #2: Yes

3. Has the statistical analysis been performed appropriately and rigorously? 

Reviewer #1: Yes

Reviewer #2: Yes

4. Have the authors made all data underlying the findings in their manuscript fully available?

Reviewer #1: Yes

Reviewer #2: Yes

5. Is the manuscript presented in an intelligible fashion and written in standard English?

Reviewer #1: Yes

Reviewer #2: Yes

6. Review Comments to the Author

Reviewer #1: 1. I appreciate that the authors saw my concerns here and agreed this is worthy of addressing. The analysis which was run however isn’t exactly what I had in mind, which may be due to a lack of clarity on my part. The more appropriate analysis would compare iterative models —rating ~ outcome vs. rating ~ outcome + accuracy vs. rating ~ outcome + accuracy + difficulty. The crucial step here is comparing the last two models as it allows you to assess whether adding difficulty improves the model fit above accuracy alone. This is roughly equivalent to what Westbrook et al. (2013) + others have done.

While I don’t think not doing this precludes publication, I think the authors should consider this approach and include it in the main manuscript (or at least supporting information) if they agree. The current approach in the Supporting Information is OK, but it doesn’t allow you to assess whether difficulty adds anything over and above accuracy. The authors, however, are of course correct in saying the current experiment doesn’t allow this issue to be resolved entirely and it’s a good suggestion to note this in the General Discussion, as they have.

I believe all the other amendments to both mine and Reviewer 2’s suggestions are appropriate and have hopefully improved the paper!

Good work again on an interesting pair of experiments. I’ve enjoyed reading this work.

Reviewer #2: I thank the authors for thoroughly addressing all of my comments. I have no further concerns and I am happy to recommend this manuscript for publication.

7. PLOS authors have the option to publish the peer review history of their article (what does this mean?). If published, this will include your full peer review and any attached files.

Reviewer #1: **Yes: **Jake Embrey

Reviewer #2: **Yes: **Mario Bogdanov

---

## [Author Response · Author response to Decision Letter 1]

25 Oct 2023

Please refer to Response to Reviewers.pdf for a better formatting.

PONE-D-23-18706

Need for cognition moderates the relief of avoiding cognitive effort PLOS ONE

Reviewer 1

I appreciate that the authors saw my concerns here and agreed this is worthy of addressing. The analysis which was run however isn’t exactly what I had in mind, which may be due to a lack of clarity on my part. The more appropriate analysis

would compare iterative models —rating ~ outcome vs. rating ~ outcome + accuracy vs. rating ~ outcome + accuracy + difficulty. The crucial step here is comparing the last two models as it allows you to assess whether adding difficulty improves the model fit above accuracy alone. This is roughly equivalent to what

Westbrook et al. (2013) + others have done.

While I don’t think not doing this precludes publication, I think the authors should consider this approach and include it in the main manuscript (or at least supporting information) if they agree. The current approach in the Supporting Information is OK, but it doesn’t allow you to assess whether difficulty adds anything over and above accuracy. The authors, however, are of course correct in saying the current experiment doesn’t allow this issue to be resolved entirely and it’s a good

suggestion to note this in the General Discussion, as they have.

 I believe all the other amendments to both mine and Reviewer 2’s suggestions are

 appropriate and have hopefully improved the paper!

 Good work again on an interesting pair of experiments. I’ve enjoyed reading this

work.

Response:

We thank the reviewer for his positive evaluation of our revised manuscript, and for wanting to further clarify how to best approach this analysis. Indeed, comparing iterative models including additional predictors, keeping every other predictor equal, should allow to verify the existence of additional explanatory power of the added predictors.

We implemented the suggestion of the reviewer, but rather than adding the simple main effects of outcome, accuracy, and difficulty iteratively, we decided to compare a model including the main effects and interaction of outcome and accuracy, to a model including those same predictors and the interaction of outcome and difficulty level. We believe that this approach does capture the iterative logic suggested by the reviewer (and implemented in Westbrook et al., 2013), but is better suited to answer our research question. In fact, the main effects of difficulty level and accuracy alone do not explain much variance in the data, but they do show strong interactions with outcome.

Our results confirm the pattern of results previously reported in the Supplementary Information. For Experiment 1, a simple model including the interaction of accuracy with outcome alone performed better than the model including both interaction terms,

suggesting the interaction with difficulty level does not have explanatory power above the interaction of outcome with accuracy. Rather, the more complex model with both interaction terms performed better than one including the simple interaction with difficulty level. For Experiment 2, the opposite pattern was found, in line with the analysis reported in the previous round of revision. More specifically, the model including two interaction terms performed much better than the one including the interaction of outcome with accuracy alone. In other words, the interaction of outcome with difficulty level survived after controlling for the interaction of outcome with accuracy. We have included these results in the Supplementary Information.

We are grateful to the reviewer for pointing out this additional approach which further clarifies that, in the second experiment, there is no evidence that individual differences in arithmetic performance (and therefore, in reward expectations) confounded the relief effect of avoiding high vs. low effort.

Reviewer 2

I thank the authors for thoroughly addressing all of my comments. I have no further concerns and I am happy to recommend this manuscript for publication.

Response:

We are very grateful to the reviewer for the insightful suggestions offered in the previous round of revision, that strongly helped us improving the paper.

---

## [Editor Report · Decision Letter 2]

2 Nov 2023

Need for cognition moderates the relief of avoiding cognitive effort

PONE-D-23-18706R2

Dear Dr. Gheza,

We’re pleased to inform you that your manuscript has been judged scientifically suitable for publication and will be formally accepted for publication once it meets all outstanding technical requirements.

Kind regards,

Poppy Watson

Academic Editor

PLOS ONE
---

## [Editor Report · Acceptance letter]

8 Nov 2023

PONE-D-23-18706R2 

Need for cognition moderates the relief of avoiding cognitive effort 

Dear Dr. Gheza:

I'm pleased to inform you that your manuscript has been deemed suitable for publication in PLOS ONE. Congratulations! Your manuscript is now with our production department. 

Kind regards, 

on behalf of

Dr. Poppy Watson 

Academic Editor

PLOS ONE